# The *MIR157*–SPL15 module regulates flowering and inflorescence development in *Arabidopsis thaliana* under short days and in *Arabis alpina*

Adrian Roggen[1¤a☺], Alba Lloret[1¤b☺], Yohanna Miotto[1☺], Kang Wang[1], Kerstin Luxa[1], Vidya Oruganti[1], Serena Della Pina[1¤c], Annabel D. van Driel[1¤d], Youbong Hyun[1¤e], Bruno Huettel[2], George Coupland[1*]

1 Department of Plant Developmental Biology, Max Planck Institute for Plant Breeding Research, Cologne, Germany, 2 Max Planck Genome Centre Cologne, Max Planck Institute for Plant Breeding Research, Cologne, Germany

¤a Current address: Deleplanque & Cie, Maisons-Laffitte, France
¤b Current address: Instituto de Biología Molecular y Celular de Plantas, Valencia, Spain
¤c Current address: IQVIA, Amsterdam, The Netherlands
¤d Current address: Laboratory of Plant Physiology, Wageningen University and Research, Wageningen, The Netherlands.
¤e Current address: Laboratory of Plant Systems Evolution, Seoul National University, Seoul, South Korea
☺ The authors contributed equally to this work.
* coupland@mpipz.mpg.de

## Abstract

The plant life cycle progresses through distinct phases defined by the morphology of the organs formed on the shoot. In Arabidopsis, age-dependent reduction in the related microRNAs miR156 and miR157 controls transitions from juvenile to adult vegetative phase and from adult to reproductive phase. However, whether these miRNA isoforms have specific contributions remains unclear. To compare their roles, we used Trans-kingdom, rapid, affordable Purification of RISCs (TraPR) for small RNA sequencing, CRISPR-Cas9, and confocal imaging. We show that in shoot apices, levels of miR156 in RNA-induced silencing complexes (RISCs) decline more rapidly than those of miR157, so that miR157 is more abundant than miR156 in RISCs of older plants undergoing floral transition and inflorescence development. Accordingly, confocal microscopy analysis showed that *MIR156A* and *MIR156C* are not detectably expressed in shoot apices of older plants, whereas at this stage *MIR157C* is expressed in upper stems, and *MIR157D* is expressed in axils of inflorescence leaves. Arabidopsis flowers much earlier under long days (LDs) than short days (SDs). CRISPR-induced *mir157c* mutations but not *mir156ac* mutations accelerated flowering under SDs, and altered inflorescence leaf morphology. Notably, *mir157c* mutations also caused early flowering in *Arabis alpina*, a perennial relative of Arabidopsis, indicating that the repression of flowering by this paralogue is evolutionarily conserved. SPL15 transcription factor promotes flowering under SDs and its mRNA is a target of miR156/miR157. SPL15 abundance was higher in apices of *mir157 cd* mutants under SDs, and *spl15* mutations

**Data availability statement:** Small RNA-sequence reads are available in NCBI SRA database (BioProjectID: PRJNA1096355). The numerical data underlying the graphs can be found in S2 Table. George Coupland is responsible for distributing the genetic materials generated for this publication.

**Funding:** This work was supported by a grant from the Deutsche Forschungsgemeinschaft (https://www.dfg.de/, CO318/11-1) and by the DFG-funded Cluster of Excellence (https://www.dfg.de/, EXC2048/1 Project ID: 390686111) to GC, and the laboratory of GC receives core funding from the Max Planck Society. This work was also supported by postdoctoral fellowships from the Alexander von Humboldt Stiftung to AL. YM was supported by the DFG-funded Cluster of Excellence (https://www.dfg.de/, EXC2048/1 Project ID: 390686111) and the von Humboldt Foundation. The funders had no role in study design, data collection and analysis, decision to publish, or preparation of the manuscript.

**Competing interests:** The authors have declared that no competing interests exist.

partially suppressed the early flowering of *mir157c* mutants and this effect was enhanced by *spl4* mutation. We show by genetic analysis that the florigen FLOWERING LOCUS T overcomes the requirement for SPL15 in LDs but not SDs, contributing to the increased importance of the *MIR157C–SPL15* module under SDs. We conclude that *MIR157* genes have important evolutionarily conserved roles in repressing floral transition and modulating inflorescence development of older plants under SDs.

## Author summary

In plants, microRNA156 (miR156) and related miRNAs regulate phase transitions in the life cycle by negatively regulating SQUAMOSA PROMOTER-BINDING ROTEIN-LIKE (SPL) transcription factors (TFs). We compare the roles of miR156 and miR157 in the transition from vegetative development to flowering in Arabidopsis using Trans-kingdom, rapid, affordable Purification of RISCs for small RNA sequencing, CRISPR-Cas9 reverse genetics, and confocal imaging. The abundance of miR156 and miR157 is higher in juvenile plants where they inhibit the progression to adult phase. We find, as previously reported, that two of the eight genes encoding miR156 are expressed in this characteristic age-dependent pattern in apices, but that expression of *MIR156B*, *MIR157C* and *MIR157D* persists, or occurs in precise spatial patterns, during flowering. Particularly, *MIR157C* negatively regulates flowering under short days, where flowering of wild-type is delayed and SPL TFs control flowering time. We show that the role of *MIR157C* in negatively regulating flowering is conserved in perennial *Arabis alpina*, and that in Arabidopsis it negatively regulates SPL15 at the shoot apex. Moreover, we genetically dissect the daylength-dependent role of SPL15 in flowering and inflorescence development. Our data highlight the importance of the *MIR157C–SPL15* module in controlling floral transition under short days.

## Introduction

The plant life cycle can be characterized as a series of transitions between different developmental phases that are controlled by endogenous and environmental cues [1,2]. After seed germination, the life cycle progresses through juvenile vegetative, adult vegetative, and reproductive phases [2,3]. These different phases can be distinguished by morphological features that differ among species, but usually include leaf shape and size, and the identity of the organs formed at the shoot apical meristem (SAM) [2–4]. In Arabidopsis, floral transition occurs during the transition from the adult vegetative phase to the reproductive phase. The related microRNAs (miRs) miR156 and miR157 and their *SQUAMOSA PROMOTER-BINDING PROTEIN-LIKE* (*SPL*) target genes play important roles in regulating the rate of transition through developmental phases [2,3,5,6]. Both miR156 and miR157 are complementary to *SPL* transcripts with a single mismatch, although the sequence of the predominant isoforms differs by three nucleotides [7–9].

The levels of these miRNAs are relatively high in cotyledons and leaves formed on younger plants, and decrease to lower levels in leaves formed later during shoot development and in older apices [10–12]. This age-related pattern of miR156/miR157 abundance and the antagonistic interaction between miR156/miR157 and SPL transcription factors (TFs) serves as a developmental timer during vegetative growth, conferring age-related differences in leaf development [11,13,14], and providing a basis for age-dependent flowering [10,12,15,16]. Recently, Arabidopsis plants with mutations in all 12 genes encoding miR156/miR157 were described, and the complete loss of miR156/miR157 had profound effects on plant development, causing premature adult leaf morphology and rapid transition to flowering [17]. Four out of these 12 genes (*MIR156A, MIR156C, MIR157A* and *MIR157C*) express approximately 90% of mature miR156/miR157 in 11-day-old seedlings [9]. Consistent with the phenotypes of *miR156/miR157* mutants, transgenic plants that express miRNA156/157-resistant versions of *SPL* gene transcripts (*rSPL*), such as *rSPL3*, *rSPL9* and *rSPL15*, show precocious acquisition of adult leaf morphology and early flowering [10–12,15,16].

The transition to flowering in Arabidopsis is rapidly initiated in response to long days (LDs) and can be divided into two further phases [1,18–20]. First, the vegetative SAM transitions to an inflorescence meristem (IM) that forms cauline leaves with inflorescence branches in their axils, concurrent with rapid shoot elongation (I1 phase). Subsequently, the IM initiates the formation of floral primordia and the outgrowth of leaves is suppressed (I2 phase). Under LDs, both I1 and I2 occur rapidly through the activity of the photoperiodic pathway that activates transcription of *FLOWERING LOCUS T* (*FT*) and *TWIN SISTER OF FT* (*TSF*) in phloem companion cells, and the FT and TSF proteins move to the SAM to induce floral transition [1,21–23]. During I2, the stable expression of floral meristem identity genes such as *LEAFY* (*LFY*) and *APETALA1* (*AP1*) in emerging floral primordia marks the initiation of continuous flower production in the inflorescence and thus completes the floral transition [24,25]. The role of SPL TFs in flowering is most important under short days (SDs) when flowering occurs more slowly than under LDs, and *FT* and *TSF* are not expressed [1,12,15]. Accordingly, mutations in *MIR156* and *MIR157* genes do not affect flowering time when plants are exposed to LDs even for short lengths of time [26]. Eleven SPL TFs are repressed by miR156 and miR157 [2,3,5], and of these SPL15 has an important role in promoting floral transition under SDs, and *SPL3*, *4*, *5* and *9* also contribute [12,15,27–30].

Mutations in *SPL15* cause late flowering under SDs, and SPL15 protein accumulates at the SAM prior to floral transition under these conditions [15]. Moreover, SPL15 accumulates in the SAM earlier in development when expressed from an *rSPL15* transgene that is insensitive to miR156/miR157 rather than from wildtype *SPL15* and promotes earlier flowering, suggesting that the decline in miR156/miR157 during development allows the accumulation of SPL15 at the SAM of wild-type plants. SPL15 then interacts with the MADS-box transcription factor SUPPRESSOR OF OVEREXPRESSION OF CONSTANS1 to activate transcription of *FRUITFULL* (*FUL*) and *MIR172B* [15]. FUL and miR172 accelerate floral transition by reducing expression of APETALA2-LIKE (AP2-LIKE) TFs in the SAM at the transcriptional and post-transcriptional levels, respectively [31–34]. In response to LDs, both *FUL* and *MIR172B* are rapidly expressed, and this probably occurs largely independently of SPL15 through the FT TSF system [31,32,35,36]. Regulation of flowering by the miR156/miR157–SPL module is conserved in perennial relatives of Arabidopsis such as *Arabis alpina* and *Cardamine flexuosa,* and strongly influences the age-related flowering response to winter temperatures (vernalization) that occurs under SDs [37–39]. In these species, miR156/miR157 have an important role in repressing vernalization response in young plants by inhibiting SPL15 expression.

Although previous studies have shown that miR156/miR157 are negative regulators of vegetative phase change and floral transition, the relative contribution of these two isoforms and the paralogues that encode them is unclear. Sequence analysis of small RNAs in different tissues indicated that both miR156 and miR157 are expressed in juvenile seedlings, but that miR157 is expressed at a higher level in inflorescence tissues than miR156 [9,17], suggesting that the isoforms might have different functions at specific stages of development. In this study, we describe the spatio-temporal expression patterns of several paralogues encoding miR156 or miR157, and genetically analyse those that are most highly expressed to dissect their roles in repressing flowering and in inflorescence development. We demonstrate that during plant development the level of miR157 in RNA-induced silencing complexes (RISCs) declines at the shoot apex more slowly than that of miR156, and that miR157 is the more important negative regulator of floral transition and inflorescence development

under non-inductive SDs when SPL TFs most strongly regulate flowering. Moreover, *MIR157C* and *MIR157D* show specific patterns of expression in the stem and cauline leaf axils during and after floral transition. We show that they negatively regulate SPL15 to repress flowering, and that SPL15 is important in controlling the vegetative to I1 transition when *FT TSF* are inactive under SDs or in *ft tsf* mutants. Therefore, we find that although miR156 levels decline steeply with age, *MIR157* genes strongly influence traits that are expressed late in plant development such as flowering time under SDs and inflorescence development, and their role in flowering time is conserved in related perennial plants.

## Results

### Time-resolved small RNA sequencing reveals different dynamics of miR156 and miR157 within RISCs in apices

To provide a quantitative analysis of the dynamics of miR156 and miR157 levels in shoot apices during shoot development and floral transition, time-resolved small RNA (sRNA) sequencing was performed on sRNAs purified using Trans-kingdom, rapid, affordable Purification of RISCs (TraPR) [40]. This approach enriches for sRNAs loaded into RNA-induced silencing complexes (RISCs). The importance of the miR156/miR157–SPL module on flowering differs between SD and LD conditions [12,15]; therefore, miRNA levels were compared in wild-type plants grown under both daylengths (Fig 1A, B).

Shoot apices from late-flowering SD-grown plants were harvested weekly between 1 and 6 weeks after germination, whereas samples from early-flowering LD-grown plants were collected 7, 11, 14 and 18 days after germination. After TraPR purification and sRNA sequencing, the sRNA reads were mapped to the genome and counted (see Material and Methods). Most *MIR156* paralogues encode an identical miR156 isoform, and *MIR157A* to *MIR157C* encode the same miR157 isoform [41]; therefore, miRNA abundance was analysed for miR156 and miR157 separately, but not for each individual paralogue.

In the LD and SD time courses, the levels of miR156 in RISCs in shoot apices were highest at the first time point and decreased steeply afterwards (Fig 1A, B). At 7LDs, the level of miR156 was on average twice as high as that of miR157 (Fig 1A and S1 Table). However, at the 11LD and 14LD time points, both miRNAs were approximately equally abundant. After 18LDs, Col-0 plants have undergone floral transition and formed floral primordia [42], and at this time point the miR156 level decreased to 60% of that of miR157 (S1 Table). Thus, in apices of LD-grown plants, miR156 was more abundant in RISCs than miR157 during early vegetative growth but decreased to a similar and then to a lower level than that of miR157 in mature inflorescence apices.

In apices of SD-grown plants, miR156 was initially four times more abundant in RISCs than miR157 (1wSD, S1 Table), but miR156 levels decreased steeply between 1wSD and 2wSD, whereas the abundance of miR157 showed a more moderate decrease (Fig 1B). Consequently, miR156 and miR157 were approximately equally abundant in apices of 2-week-old SD-grown plants, and subsequently the levels of miR157 were consistently higher than those of miR156 (S1 Table). At 5wSD and 6wSD, when Col-0 plants initiate floral transition, mature miR156 levels were approximately one-third of those of miR157. We conclude that although miR156 and miR157 levels in RISCs in apices decrease as the plants age, they are still present at the shoot apex around the time of floral transition (14LDs or 6wSDs) and in the early inflorescence meristem, and that particularly under SDs, miR157 is more abundant in RISCs at floral transition than miR156.

To determine the contributions of miR156 and miR157 to flowering time and inflorescence development, CRISPR-Cas9 reverse genetics was used to recover mutant alleles of *MIR156* and *MIR157* genes (see Material and Methods). First, the levels of all *MIR156* and *MIR157* precursor RNAs in apices during vegetative development and early inflorescence development were assessed under LDs using RNAseq data [42] (S1A Fig). As previously described, *MIR156A* and *MIR156C* genes are the most highly expressed *MIR156* genes [9] and are reduced in expression in apices as plants age [13]; therefore, we introduced new mutations into these genes to increase the allelic variation at these loci [17] and combined the mutations in a double mutant. *MIR156D* is also relatively highly expressed in apices, but no mutant allele could be recovered for this gene. Novel deletion alleles were generated in all *MIR157* genes (*MIR157A*, *MIR157B*, *MIR157C* and *MIR157D*; S1B Fig). The *mir157a-3 mir157b-3 mir157c-3* triple mutant was then constructed to assess the contribution of

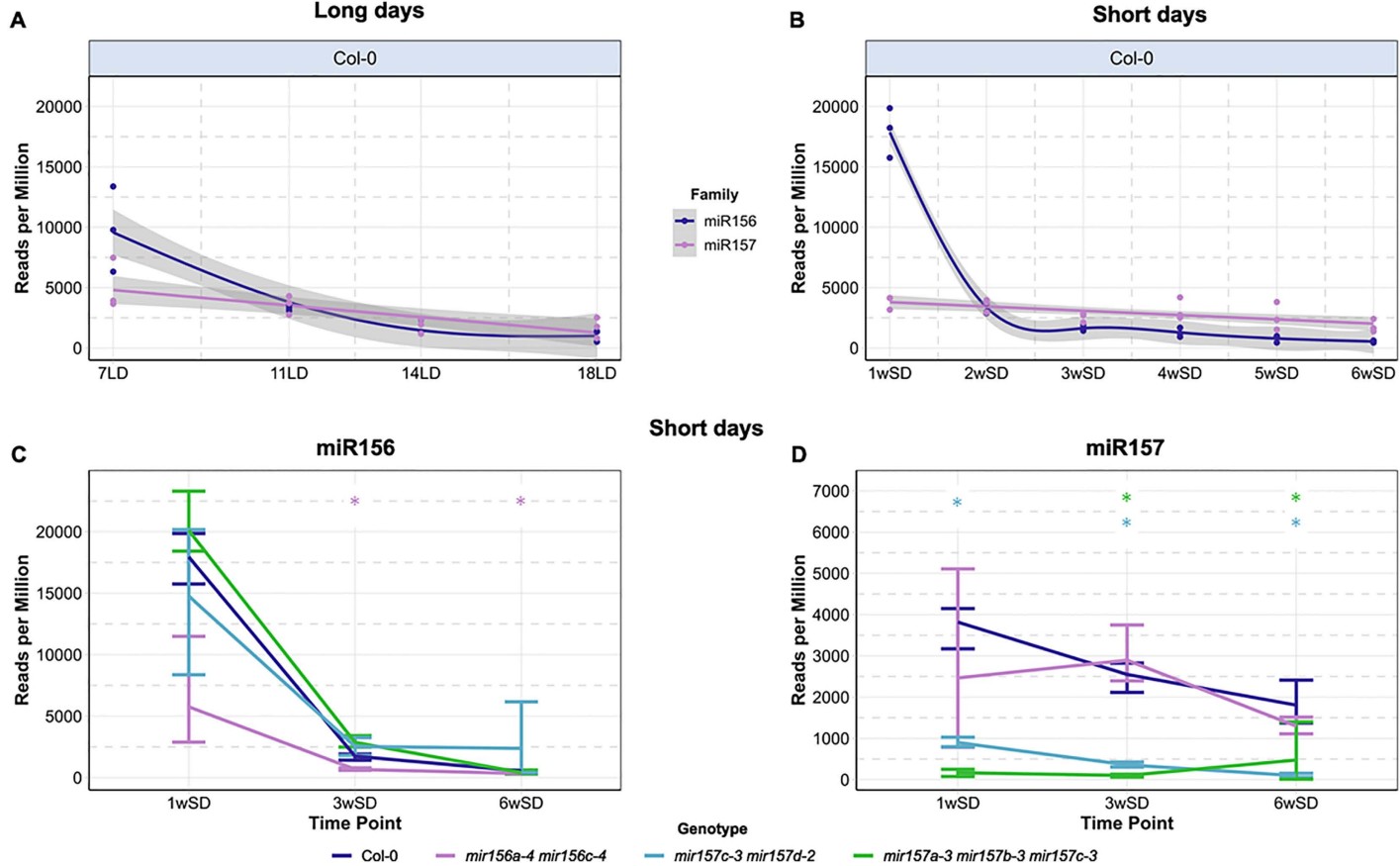

**Fig 1. Time-resolved small RNA-sequencing reveals different expression dynamics of miR156 and miR157 families.** The abundance of miR156 and miR157 isoforms in apices of wild-type (Col-0) plants in (A) long-day (LD) and (B) short-day (SD) time courses. Material was harvested at the indicated time points. Based on microscopic analysis of plants under the same growth conditions, the 7LD and 11LD samples are vegetative apices, 14 LD is during the I1 to I2 transition, and 18LD samples are I2 [33]. The 1wSD to 5wSD samples represent the extended vegetative phase under SDs, whereas the 6wSD samples are I1 apices (Fig 4A). Lines represent fitted generalised additive models (see Methods). Shaded areas represent approximate 95% confidence intervals. The expression of miR156 isoforms (C) and miR157 isoforms (D) in Col-0 (dark blue), *mir156a-4 mir156c-4* (purple), *mir157c-3 mir157d-2* (light blue), and *mir157a-3 mir157b-3 mir157c-3* (green) mutants. Data points represent means of three independent biological replicates. Error bars represent standard deviation. Asterisks in (C) and (D) represent significance differences in miR156 or miR157 levels between mutant and Col-0 at a given time point as determined by pairwise Wilcoxon *t*-tests at $p < 0.05$.

*MIR157D* alone, which appeared to be only expressed during inflorescence development (16LD; S1A Fig). Similarly, the *mir157c-3 mir157d-2* double mutant was constructed to assess the effect of inactivating the two *MIR157* genes mainly expressed during inflorescence development. To assess the reduction in the abundance of miR156 and miR157 caused by these combinations of mutations, sRNA sequencing was performed after carrying out TraPR purification on apical samples of *mir156a-4 mir156c-4, mir157c-3 mir157d-2* and *mir157a-3 mir157b-3 mir157c-3* mutants grown from vegetative development (1 and 3 weeks under SDs) to floral transition (6 weeks under SDs) (Fig 1C, D). The levels of the miRNAs in RISCs at each time point were then compared with those of Col-0 and the other mutants (Fig 1C, D). At 1wSD, when its expression was highest in all genotypes, miR156 levels in *mir156a-4 mir156c-4* were approximately one-third of those in Col-0 and were extremely low at later time points (Fig 1C and S1 Table). By contrast, the abundance of miR156 in *mir157c-3 mir157d-2* and *mir157a-3 mir157b-3 mir157c-3* was comparable to that of Col-0 at all three time points (Fig 1C). The miR157 levels in *mir157c-3 mir157d-2* were reduced to approximately 25% of Col-0 levels early in development

(1wSD) and to 3% of Col-0 at floral transition (6wSD), consistent with these two genes contributing the vast majority of miR157 at shoot apices during floral transition and inflorescence development (Fig 1D and S1 Table). In *mir157a-3 mir157b-3 mir157c-3*, miR157 levels were very low during vegetative development, but clearly increased at floral transition, consistent with *MIR157D* being increased in expression at this stage (Fig 1C, D and S1 Table). Therefore, the new CRISPR-Cas9-induced deletions in *MIR156* and *MIR157* genes effectively reduce miR156 or miR157 abundance in RISCs in a family-specific manner, and support the idea that *MIR157C* and *MIR157D* are mainly expressed later in development during floral transition and early inflorescence development.

## *miR156* and *mir157* mutations differentially affect vegetative and cauline leaf development

The morphology of rosette leaves changes acropetally along the main shoot axis as a consequence of vegetative phase change, and these differences are regulated by miR156/miR157 [9,11,43]. Therefore, whether *mir157* mutations have a stronger effect on later formed leaves, consistent with the miRNA sequencing results, was tested. The effects of the *mir156* and *mir157* deletion alleles on leaves were determined by calculating the length-to-width ratio (LWR) of all rosette leaves that are formed during the vegetative phase of Col-0 and mutant plants at 4wSD (Fig 2A, B and S2A, B Fig). For Col-0, the LWR of rosette leaves increased progressively from the second to the sixth leaf, where it reached its maximum (Fig 2A). The leaves formed later on the shoot had not completely expanded at 4wSDs and therefore showed a lower LWR. The LWR of the cotyledons and the first two leaves was greater in *rSPL15* plants than in Col-0 (Fig 2A and S2A, B Fig), and the fifth leaf of *rSPL15* plants had already reached a maximum LWR, consistent with an accelerated transition to the adult vegetative phase compared with Col-0. The rosette leaf phenotypes of *mir156a-4 mir156c-4* double mutants were most similar to those of *rSPL15* plants, and the LWR of the first two leaves was much higher than that of Col-0 and close to the maximum value. Therefore, *MIR156AC* play a major role in conferring juvenile vegetative phase represented by leaf morphology, as previously described [9,44,45]. A more irregular phyllotaxy compared with Col-0 was also observed for *mir156a-4 mir156c-4* (Fig 2A and S2C Fig). The variation in divergence angle between successive rosette leaves was similarly high in *rSPL15* plants (S2C Fig). In contrast to *mir156a-4 mir156c-4*, the increase in the LWR of successive leaves of *mir157c-3 mir157d-2* was largely comparable to that of Col-0 plants, but higher maximum LWR values were achieved for leaves 6, 7 and 8 (Fig 2A). Moreover, *mir157c-3 mir157d-2* and *mir157a-3 mir157b-3 mir157c-3* mutants showed no obvious defect in phyllotaxis (S2A, C Fig). Increased miR156 expression and reduced SPL activity also cause a reduction in plastochron length [46]. Therefore, we tested for differences in the rate of leaf initiation during vegetative growth under SDs. The *mir156a-4 mir156c-4* plants showed a lower leaf initiation rate (longer plastochron), consistent with increased SPL activity, whereas *mir157c-3 mir157d-2* plants unexpectedly appeared to produce leaves faster than that of Col-0 plants (Fig 2C). Collectively, these results indicate that *MIR156AC* are important early during shoot development to confer juvenile vegetative phase, but that *MIR157* genes have a more marked function in later rosette leaves, consistent with miR157 being more abundant than miR156 in RISCs in older apices.

We therefore assessed the effects of miR156 and miR157 on the shape and number of cauline leaves formed in the inflorescence during the I1 phase. The *mir157c-3* and *mir157d-2* mutants both formed significantly more cauline leaves than Col-0, particularly under SDs (Fig 2D and S2 Table), and the double mutant showed an additive phenotype (Fig 2D and S2 Table). The triple mutant *mir157a-3 mir157b-3 mir157c-3* formed fewer cauline leaves than *mir157c-3 mir157d-2* (S2 Table). Moreover, *mir156a-4 mir156c-4* mutants produced the same number of cauline leaves as Col-0 (Fig 2E), consistent with *MIR156AC* having a smaller effect than *MIR157 CD* genes on inflorescence development. The morphology of the cauline leaves of *mir157c-3*, *mir157d-2* and *mir157c-3 mir157d-2* mutants also differed from Col-0 (Fig 2F, G). The cauline leaves of these mutants were more serrated in terms of the number and the depth of the serrations than those of Col-0 and *mir156a-4 mir156c-4*. Moreover, the LWR of the cauline leaves was lower in *mir157d-2* than Col-0, and unexpectedly this effect was suppressed in the mir157c-3 mir157d-2 double mutant (Fig 2G). Taken together, this

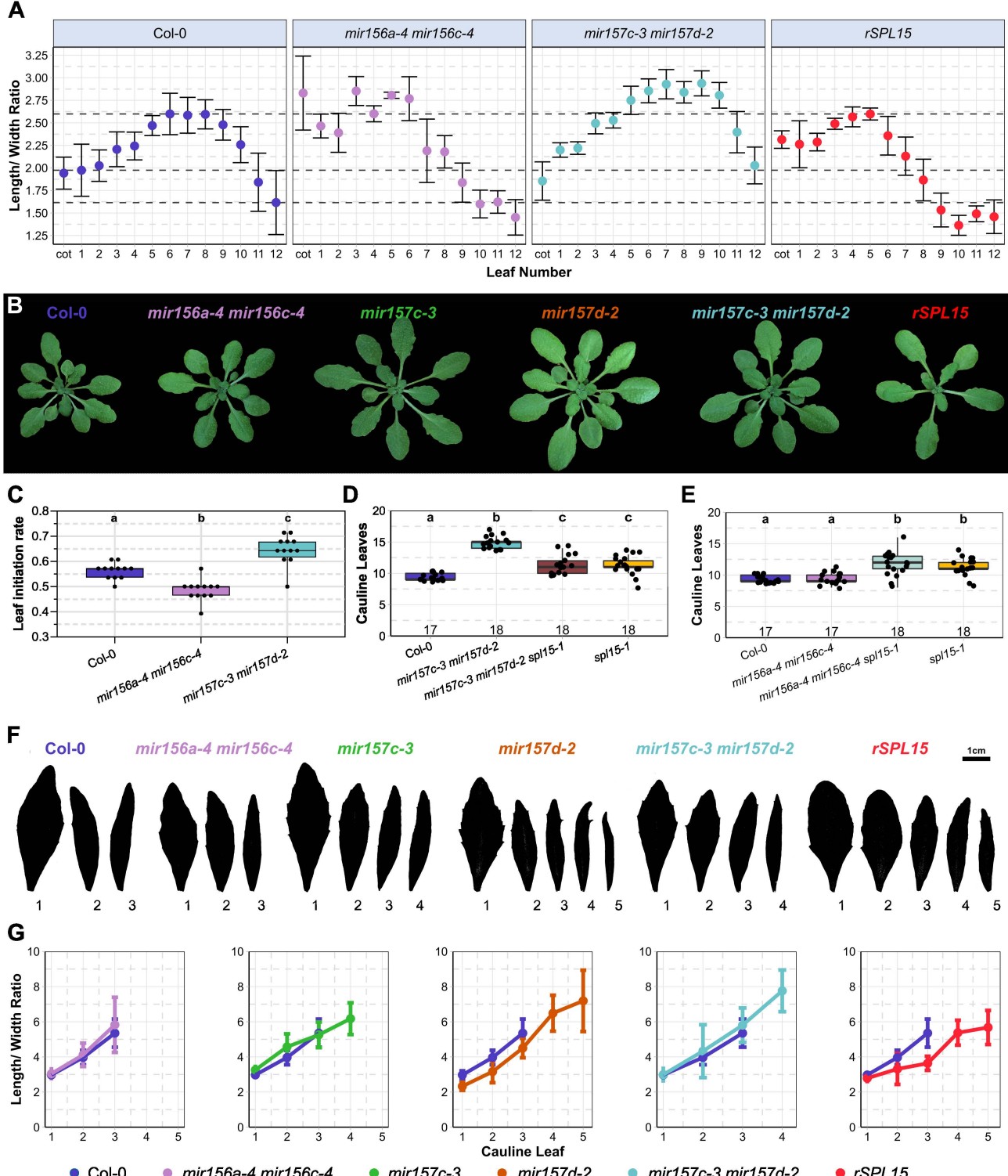

**Fig 2. Analysis of *mir156* and *mir157* mutants reveals different roles for mir156 and mir157 in the regulation of leaf number and shape.** (A) The length/width ratio of cotyledons and rosette leaves after growth for 4 weeks in short days (4wSD). Dashed lines show the minimum and maximum mean values for Col-0. (B) Rosette phenotype of Col-0, *mir156a-4 mir156c-4*, *mir157c-3, mir157d-2*, *mir157c-3 mir157d-2* and *rSPL15* plants grown for 4wSD. (C) Leaf initiation rate of each genotype defined as number of leaves formed per day and measured at 4-weeks short days. (D–E) The number of

analysis demonstrates that *MIR157 CD* genes are more important in regulating cauline leaf number and morphology than *MIR156AC*.

*SPL15* is one of the miR156/miR157 targets that regulates floral transition under SDs [15], and therefore the effect of SPL15 on the enhanced cauline leaf number of *mir157c-3 mir157d-2* was tested. Under SDs, *spl15–1* (11.33 cauline leaves; Fig 2D and S2 Table) was epistatic to *mir157c-3 mir157d-2* (14.94 cauline leaves) with respect to cauline leaf number in the *spl15–1 mir157c-3 mir157d-2* triple mutant (11.28 cauline leaves), suggesting that increased SPL15 expression might confer this phenotype. Overall, consistent with the sRNA sequencing data, analysis of leaf morphology suggests that *MIR156AC* are important for regulating juvenile vegetative leaf morphology and phyllotaxis, whereas *MIR157 CD* are more important in controlling cauline leaf number and shape in the inflorescence.

### *mir156a-4 mir156c-4* and *mir157c-3 mir157d-2* mutants differ in their flowering responses to daylength

The SPL TFs and miR156/miR157 regulate floral transition [10,12,15]; therefore, the flowering time of *mir156a-4 mir156c-4* and *mir157c-3 mir157d-2* mutants was scored under LDs and SDs, and compared with that of Col-0 and *rSPL15* plants. Overexpression of *MIR156* genes or mutation of *SPL* genes caused a shorter plastochron than in Col-0 [46,47], therefore, we concentrated on measuring flowering time by assessing the number of days until the inflorescence stem started to elongate above the rosette (bolting time, see Material and Methods). However, rosette leaf number, cauline leaf number, days to first open flower and bolting time are provided for all genotypes in S2 Table. Under LDs and SDs, *rSPL15* plants bolted significantly earlier than Col-0, but the difference was more extreme under SDs in which bolting of Col-0 was delayed (Fig 3A, B and S2 Table). The *mir157c-3 mir157d-2* double mutant also consistently bolted earlier than Col-0 under SDs, whereas the *mir156a-4 mir156c-4* plants did not (Fig 3B). Therefore, *MIR157 CD* play a more important role than *MIR156AC* in repressing bolting time under SDs. By contrast, under LDs, *mir157c-3 mir157d-2* mutants bolted at the same time as Col-0 (Fig 3A), whereas the bolting time of *mir156a-4 mir156c-4* relative to Col-0 varied among experiments. The *mir156a-4 mir156c-4* mutant bolted significantly earlier than Col-0 under LDs in some experiments (Fig 3A), whereas in others the bolting time of these genotypes did not differ (S2 Table). These data demonstrate that under SDs, when plants transition to flowering later, *MIR157 CD* have a more important role than *MIR156AC* in repressing bolting, and that under LDs *MIR156AC* have a stronger effect, although this varied among experiments presumably due to redundancy among *MIR156/MIR157* genes [17].

### FUL and SPL15 contribute to the early-bolting phenotype of *mir157c-3 mir157d-2* mutants

SPL TFs activate transcription of *FUL*, which promotes the expression of *LFY* and *AP1* in floral primordia, and *AP1* mRNA acts as a molecular marker for floral development [12,28]. Therefore, the abundance of *FUL* and *AP1* mRNAs was quantified in apices of plants of different genotypes grown under LDs and SDs as markers for SPL activity and floral primordia development. Under SDs, the level of *FUL* mRNA in *rSPL15* and *mir157c-3 mir157d-2* plants was higher than in Col-0 and *mir156a-4 mir156c-4* at 5w, 6w and 7w (Fig 3D), consistent with the bolting phenotypes of these genotypes (Fig 3B). Similarly, *AP1* mRNA was more abundant in *mir157c-3 mir157d-2* and *rSPL15* than in Col-0 and *mir156a-4 mir156c-4* at 7wSD (Fig 3D). Under LDs, the differences were less pronounced, but the level of *FUL* mRNA was slightly higher in *mir156a-4 mir156c-4* compared with other genotypes at 11LDs, and *AP1* mRNA was higher in the same genotype at 14LDs (Fig 3C). Overall, the gene expression analysis supports the hypothesis that *mir157c-3 mir157d-2* and *rSPL15* cause early floral transition under SDs, whereas *mir156a-4 mir156c-4* does not.

*FUL* mRNA abundance was increased in *mir157c-3 mir157d-2*, and *ful* mutants show delayed flowering under LDs and SDs [48,49]; therefore, whether FUL contributes to the early bolting of *mir157c-3 mir157d-2* under SDs was tested by constructing the triple mutant *ful-2 mir157c-3 mir157d-2* (Fig 3G). The *ful-2* mutation was fully epistatic to *mir157c-3 mir157d-2* and *mir157c-3* with respect to bolting time, indicating that *FUL* is essential for the early bolting caused by

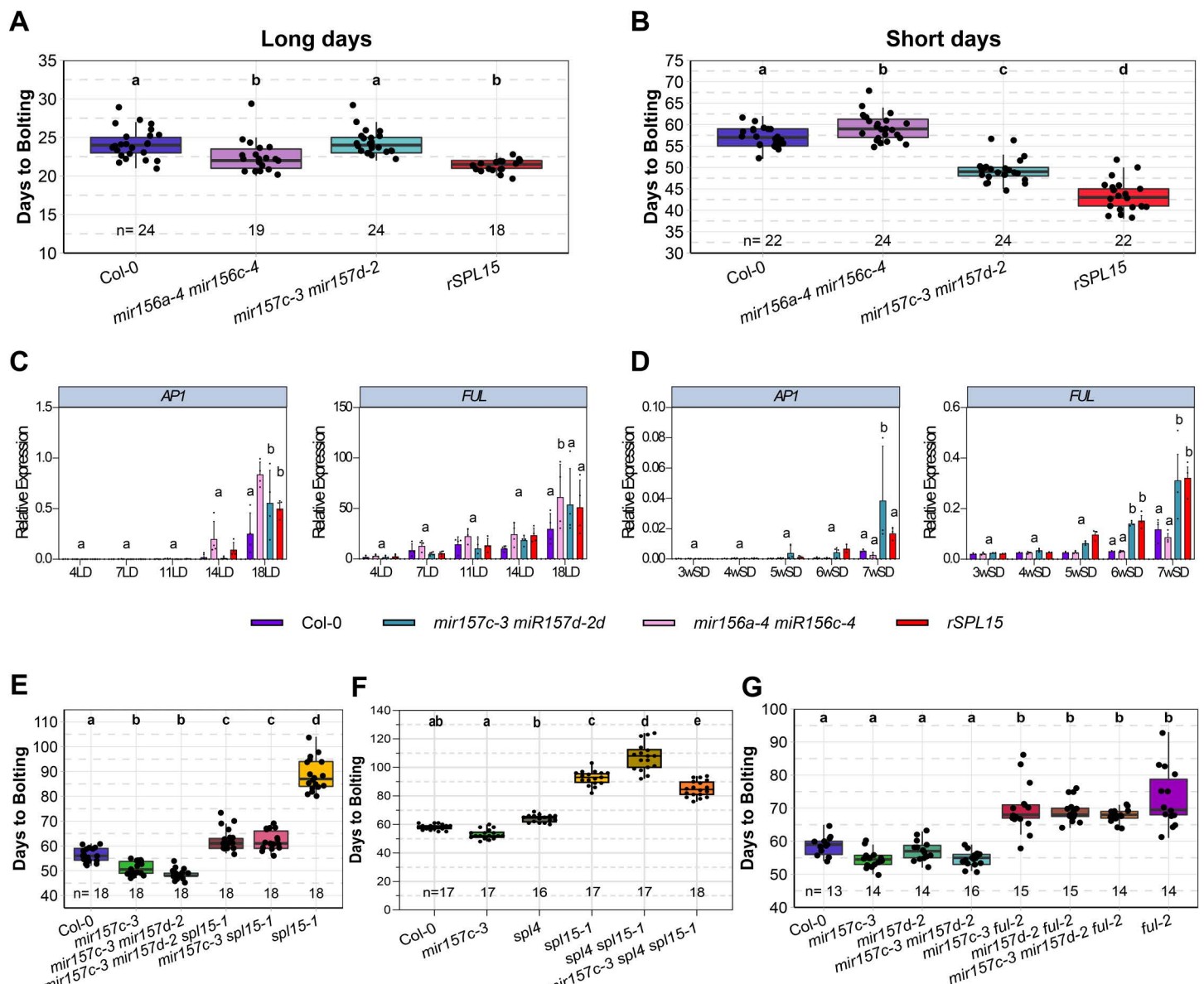

**Fig 3. Analysis of *mir156* and *mir157* mutants reveal different roles in the regulation of flowering.** (A–B) Bolting time of wild-type (Col-0) plants compared with *mir156a-4 mir156c-4*, *mir157c-3 mir157d-2* and *rSPL15* in (A) long-day (LD) and (B) short-day (SD) conditions. (C, D) *AP1* and *FUL* expression levels quantified by RT-qPCR in apices of different genotypes grown in (C) LD and (D) SD conditions. Values are means ± standard deviation. (E–G) Flowering time of the indicated genotypes under SDs expressed as the number of days between sowing and bolting. Letters indicate significant differences calculated with ANOVA and Tukey's Honest Significant Difference test; $p < 0.05$.

*mir157c-3 mir157d-2*. SPL15 activates *FUL* and loss-of-function mutations in *SPL15* delay bolting and flowering under SDs [15]; therefore, to determine whether SPL15 is required for the early-bolting phenotype of *mir157c-3 mir157d-2*, the triple mutant *spl15–1 mir157c-3 mir157d-2* was constructed. The *spl15–1 mir157c-3 mir157d-2* plants bolted later than *mir157c-3 mir157d-2* mutants and Col-0 (Fig 3E), indicating that SPL15 contributes to the early-bolting pheno-type of *mir157c-3 mir157d-2*. However, *spl15–1 mir157c-3 mir157d-2* plants bolted much earlier than *spl15–1* mutants,

cauline leaves formed by each genotype under short days. (F) Cauline leaf phenotype of Col-0, *mir156a-4 mir156c-4*, *mir157c-3*, *mir157d-2*, *mir157c-3 mir157d-2* and *rSPL15* plants grown under long days for five weeks. (G) Length/width ratio of cauline leaves compared with Col-0, leaves are shown in F. Letters indicate significant differences calculated with ANOVA and Tukey's Honest Significant Difference test; *p* < 0.05. Error bars indicate standard deviation in all panels.

suggesting that increased expression of other SPL TFs must also contribute to the *mir157c-3 mir157d-2* phenotype. The bolting times of *mir157c-3* and *spl15–1* mir157c-3 were also compared. The *mir157c-3* mutant bolted at a similar time to *mir157c-3 mir157d-2*, suggesting that the early-bolting phenotype of the double mutant is mainly caused by the *mir157c-3* mutation (Fig 3E, G and S2 Table). Moreover, the *spl15–1 mir157c-3* double mutant showed a similar bolting time to *spl15–1 mir157c-3 mir157d-2*, and bolted later than *mir157c-3* but earlier than *spl15–1* (Fig 3E). To account for the possibility that additional *SPL* genes might be involved, the *spl4 spl15–1 mir157c-3* triple mutant was generated. The *spl4 spl15–1 mir157c-3* plants bolted later than *mir157c-3* and *spl4* mutants (Fig 3F and S2 Table). However, the triple mutant plants still bolted earlier than the *spl15–1* and *spl4 spl15–1* mutants, suggesting that SPL4 and SPL15 are major contributors to the early bolting of mir157c mutants, but that other SPL genes also contribute (Fig 3F and S2 Table). Mutations in *MIR156A* and *MIR156C* did not accelerate bolting under SDs (Fig 3B and S2 Table). Nevertheless, whether *mir156a-4 mir156c-4* affected the extreme late-bolting phenotype of *spl15–1* was also tested. The triple mutant *spl15–1 mir156a-4 mir156c-4* bolted slightly earlier than *spl15–1*, but much later than Col-0 and *spl15–1 mir157c-3 mir157d-2* (S2 Table). Taken together, these results support the conclusion that *MIR157 CD* are more important in delaying floral transition under SDs than *MIR156AC* and that they do so through negatively regulating *FUL* and partially through downregulating *SPL15*.

### Spatio-temporal expression of SPL15 and FUL in *mir157 cd* mutants

The *spl15–1* mutation partially suppressed the early bolting of *mir157c-3 mir157d-2* plants; therefore, we examined whether SPL15 expression is increased in *mir157c-3 mir157d-2*. Apices were harvested from plants grown under SDs (2wSD to 6wSD; Fig 4 or under LDs (9, 11, 14 and 16LD) and analysed by confocal microscopy (see Material and Methods). First, apices of Col-0 plants carrying the *pSPL15::V9A:rSPL15* (*rSPL15*) or the *pSPL15:: V9A:SPL15* (*pSPL15*) reporter [15] were analysed for comparison under SD (Fig 4A) where the *mir157c-3 mir157d-2* early bolting phenotype was detected (Fig 3B). Consistent with previous analyses, the spatial expression patterns of *pSPL15* and *rSPL15* in the Col-0 SAM and stem cortex were similar, but *rSPL15* was expressed earlier and more strongly in both tissues (Fig 4A; 21). Then, *pSPL15* introduced into the *mir157c-3 mir157d-2* double mutant was analysed, and the fluorescence signal of VENUS9A:SPL15 was stronger in the apex of *mir157c-3 mir157d-2* than in Col-0 from at least 4wSD onwards, and was stronger in the cortex of the stem below the SAM at 5wSD and 6wSD (Fig 4A). Moreover, floral transition and elongation of the apex occurred faster in *pSPL15 mir157c-3 mir157d-2* than in *pSPL15* at 5wSD and 6wSD (Fig 4A). The pattern of VENUS9A-SPL15 was also compared in *pSPL15 mir157c-3 mir157d-2* and *pSPL15* plants grown from 9LDs to 16LDs (Fig 4B). VENUS9A-SPL15 was detected in the SAM and stem cortex in both genotypes at similar levels, consistent with *mir157c-3 mir157d-2* plants bolting at similar times to Col-0 under LDs (Fig 3A). Taken together, these data indicate that *MIR157 CD* are important in the repression of SPL15 expression in the SAM and in the stem cortex of inflorescence stem internodes under SDs. The higher SPL15 activity in *mir157c-3 mir157d-2* was also supported by increased expression of the FUL:VENUS protein in the SAM at 6wSD (Fig 4C).

### Spatial and temporal patterns of transcription of *MIR157* and *MIR156* genes at the shoot apex

The sRNA-sequencing experiment showed a progressive reduction in miR156 abundance in Col-0 apices over time, but a more constant abundance of miR157. To analyse the temporal expression patterns of individual *MIR* genes at the shoot

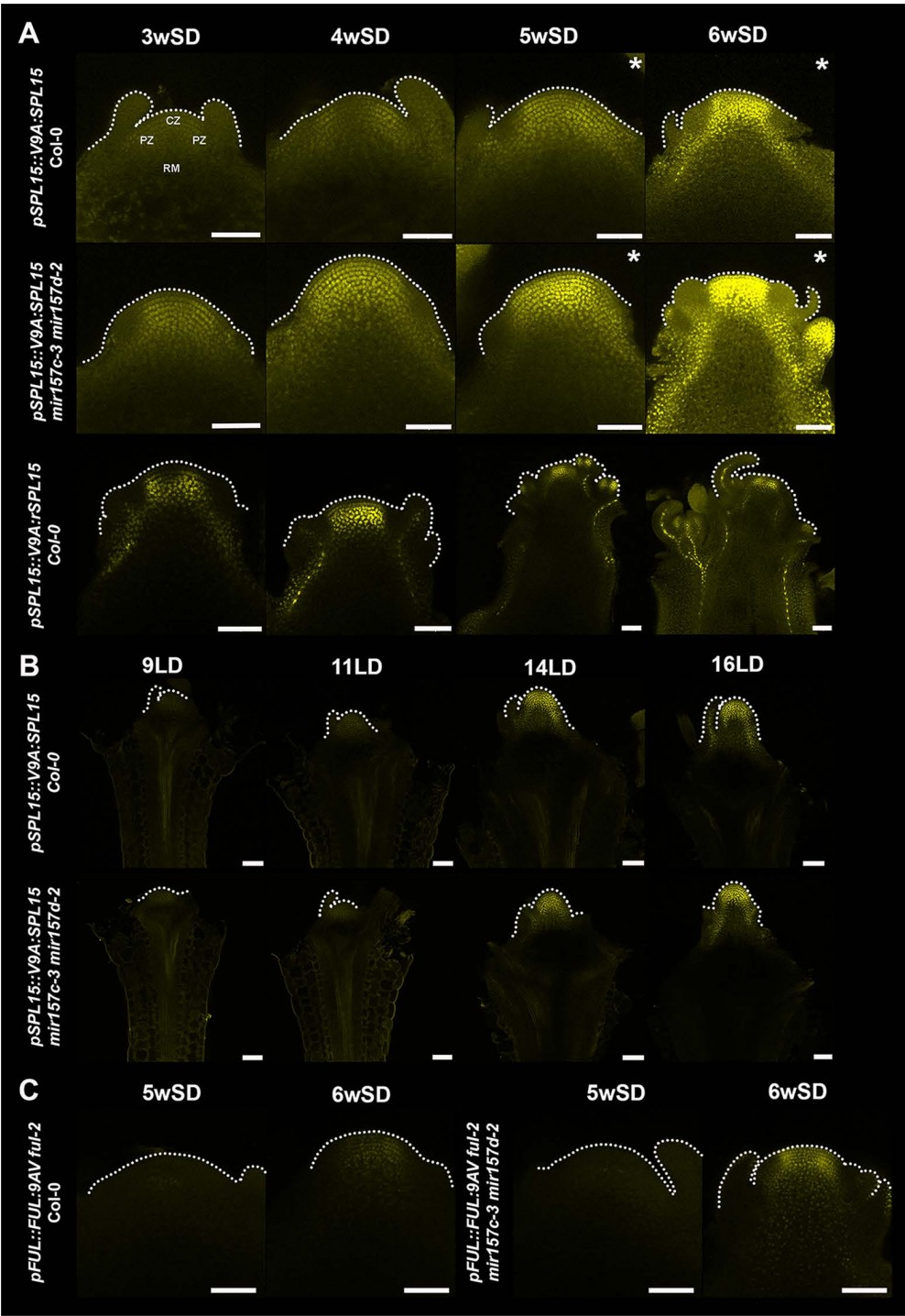

**Fig 4. Floral transition of *mir157* mutants in short days is accelerated by the premature accumulation of SPL15 and its target FUL at the shoot apical meristem.** (A) Expression of *pSPL15::V9A-SPL15* in wild type (Col-0), *mir157c-3 mir157d-2* transgenic plants, and *pSPL15::V9A-rSPL15* plants grown under SD. (B) Expression of *pSPL15::V9A-SPL15* in wild type (Col-0) and *mir157c-3 mir157d-2* transgenic plants grown under LD. (C) Confocal images of shoot apical meristems of *ful-2* and *ful-2 mir157c-3 mir157d-2* mutants expressing *pFUL::FUL: 9AV* grown under SDs. Each image is representative of five analysed samples. Fluorescence from the Venus protein is artificially colored in yellow. Asterisks mark images acquired with a lower laser intensity (see Methods). Scale bar = 50 µm. V9A, Venus-9Alanina; 9AV, 9Alanina-Venus; CZ, Central zone; RB, Rib meristem; PZ, Primordia zone.

apex, the RNA levels of the *MIR156AC* and *MIR157 CD* genes were first quantified by RT-qPCR (Fig 5A–C and S3 Fig). The levels of *MIR156A* and *MIR156C* RNA in apices of SD-grown Col-0 decreased more or less exponentially, from 1wSD until basal levels were reached after 4wSD (Fig 5A). A comparable expression pattern was also observed in LD-grown plants, where the levels of *MIR156A* and *MIR156C* decreased rapidly from 4LD to 11LD (S3A Fig), similar to changes in the level of the mature miR156 (Fig 1B) and as previously reported in seedlings [12] and apices [13]. By contrast, the levels of *MIR157A* and *MIR157C* mRNAs decreased more moderately from 1wSD and stabilised after 3wSD (S3 Fig). The *MIR157C* and *MIR157D* transcripts were therefore studied from 3wSD to 7wSD. *MIR157C* transcripts were detected at all time points and their abundance decreased by half between 4wSD and 5wSD and was subsequently maintained at a constant level (Fig 5B), whereas *MIR157D* expression was barely detectable in vegetative apices (until 6wSD), but increased when plants underwent floral transition (7wSD, Fig 5B). These results are consistent with the extremely low abundance of miR157 detected between 1wSD and 6wSD in *mir157a-3 mir157b-3 mir157c-3* mutants, in which *MIR157D* was the only functional gene encoding miR157 (Fig 1D). *MIR157C* and *MIR157D* RNAs were also detected in cauline leaves of 36LD plants (Fig 5C), demonstrating that miR157 is still present in leaves formed late in shoot development on the inflorescence and consistent with the effects of *mir157 cd* mutations on cauline leaf number and shape (Fig 2).

Transcriptional reporter lines were then constructed to characterize the spatial expression patterns of *MIR156B, MIR156C, MIR157A, MIR157C* and *MIR157D* at the shoot apex and the previously constructed *MIR156A:GFP* [17] was included for comparison. The *MIR156A::GFP* and *MIR156C::VENUS:GUS* reporters were broadly expressed throughout apices of young plants, and these spatial patterns of expression were similar in LDs and SDs (Fig 5F and S4A, B Fig). Consistent with the temporal decrease in mature miR156 levels (Fig 1A, C), *MIR156A* and *MIR156C* reporter expression decreased in apices with increasing plant age, as observed previously for *MIR156C* [13]. This decrease occurred more rapidly under LDs, in which the signals were barely detectable at 11LD and 14LD (Fig 5F), than under SDs, when the signals were clearly detectable at 14SDs but strongly attenuated at 21SDs (S4 Fig). The spatial expression pattern of *MIR156A::GFP* differed from that of *MIR156C::VENUS:GUS* at the SAM. In LDs and SDs, *MIR156C::VENUS:GUS* was detected in the SAM one week after germination, before being restricted to the more distal rib zone at later time points. By contrast, *MIR156A::GFP* was only present in the rib region at all time points under both growth conditions. Although the expression of both reporters decreased in the shoot apex and in newly formed primordia, signal persisted lower in the shoot for longer, for example at 11LD (Fig 5F). In contrast to *MIR156A::GFP* and *MIR156C::VENUS:GUS*, *MIR156B::VENUS:GUS* was expressed exclusively in the epidermis, but was absent from the epidermis of the central zone of the SAM (S5 Fig). Expression of *MIR156B::VENUS:GUS* did not appear to reduce with the age of the plant.

Consistent with the sRNA-sequencing and the RT-qPCR data, signal of *MIR157C::VENUS:GUS* expression was detected in LDs and SDs below the SAM in the stem (Figs 5D and S7A, B Fig). Expression persisted in the stem until at least 5wSD, and in LDs was still present after floral induction at 14LD (Fig 5D). At 14LD, *MIR157C::VENUS:GUS* expression was not detected in the elongating internodes representing the bolting inflorescence stem, but was present in the lower internodes (Fig 5D). This pattern together with the early-bolting phenotype of *mir157c* mutants under SD suggests that *MIR157C* acts in the stem to inhibit internode elongation and bolting under these conditions. Expression of *MIR157C::VENUS:GUS* was also observed in cauline leaves (S7B Fig), as well as in the hypocotyl epidermis (Fig 6G and S7B Fig).

The expression of the *MIR157D::VENUS:GUS* reporter was not detected in vegetative apices (Fig 5E and S7C Fig). At floral transition, *MIR157D::VENUS:GUS* expression was observed at the adaxial base of cauline leaves, in the vicinity of the axillary meristem (Fig 5E). Moreover, *MIR157A::VENUS:GUS* expression was only detected in the vasculature of young seedlings and in cell clusters below young leaf primordia, and expression was absent from the SAM (S6 Fig).

Therefore, although *MIR156AC* expression is progressively repressed at the apex during vegetative growth in SDs, *MIR157C* expression persists in the stem until floral transition, after which it is expressed in the rosette stem and cauline

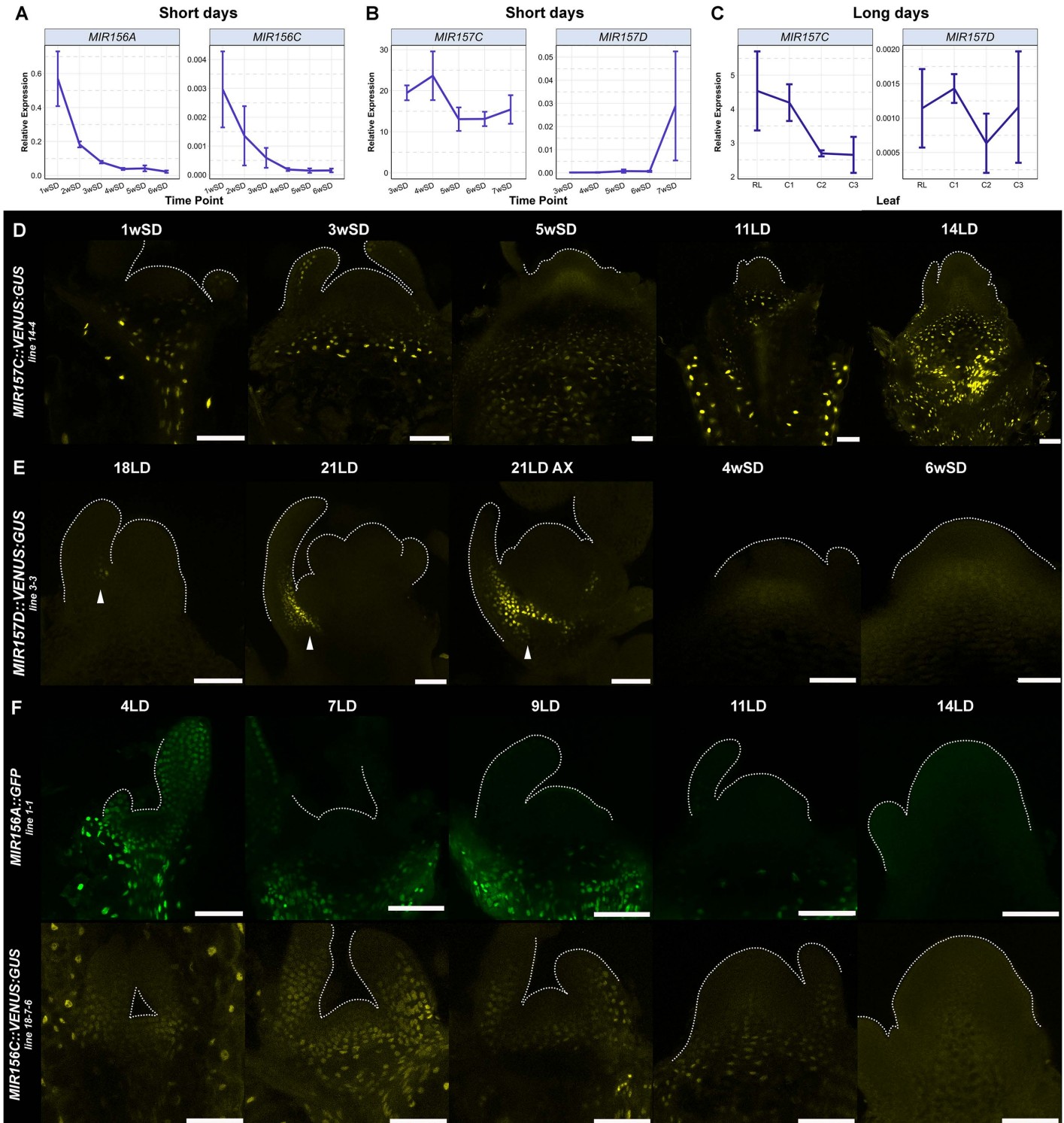

**Fig 5. Characterization of *MIR156* and *MIR157* temporal and spatial expression patterns.** (A) Expression levels of endogenous *MIR156A* and *MIR156C* genes in apices of plants grown in short days. (B) Endogenous expression of *MIR157C* and *MIR157D* genes was quantified in apices of wild-type seedlings under short days. (C) *MIR157C* and *MIR157D* expression in leaves of Col-0 plants after 36 long days. RL = youngest rosette leaf, C1 = first (lowest) cauline leaf, C2 and C3 = second and third cauline leaf respectively. Data points in panels A–C indicate the mean ± SD of three biological

replicates. (D–F) Confocal images of dissected apices of (D) *MIR157C::VENUS:GUS* in short days (SD) and long days (LDs), (E) *MIR157D::VENUS:-GUS* under LDs and SDs. Arrows indicate expression. (F) *MIR156A::GFP* and *MIR156C::VENUS:GUS* in LDs. Plants were harvested at the indicated time points after sowing. Each image is representative of five analysed samples. Fluorescence from the Venus protein is artificially coloured in yellow, fluorescence from the GFP protein is artificially coloured in green. All reporter constructs contain a nuclear localization signal. Scale bars = 50 μM.

leaves. By contrast, *MIR157D* is not expressed prior to floral induction, although it might be expressed specifically in stomatal cells [50], but later shows a characteristic expression pattern on the adaxial side and axil of cauline leaves in the inflorescence.

## Conserved function of *MIR157C* in *Arabis alpina* accelerates floral transition

In *A. alpina*, a perennial relative of Arabidopsis, the miR156–SPL module regulates the age-dependent response to vernalization [37,39]. The *mir157c-3* and *mir157d-2* alleles described above indicate that *MIR157C* has a major role in regulating bolting and flowering in Arabidopsis; therefore, we tested whether this function is conserved in *A. alpina*. First, we identified the *AaMIR156* and *AaMIR157* genes present in the *A. alpina* genome. The *MIR157* family exhibits lower conservation compared with *MIR156,* so that only two out of the four genes present in Arabidopsis (*MIR157A* and *MIR157C*) are conserved in *A. alpina* (Fig 6A). Conversely, both species retain all eight genes encoding the miR156 family. Therefore, the origin of the *MIR156* family paralogues preceded the divergence of *A. alpina* and Arabidopsis, since which time frame duplications or losses of *MIR157* family members have occurred so that *A. alpina* contains two fewer copies than Arabidopsis.

To further investigate the evolutionary relationship between *AaMIR157A* and *AaMIR157C* genes in Arabidopsis and *A. alpina*, we performed a microsynteny analysis of their respective genomic regions (Fig 6B, C). Consistent with the phylogeny (Fig 6A), a high conservation of synteny was found between the *MIR157C* regions of Arabidopsis and *A. alpina*, and the tandem duplication of *MIR157A* and *MIR157B* in Arabidopsis was present as a single copy in *A. alpina.* At the position corresponding to *MIR157D* in Arabidopsis no homologous gene was observed in the *A. alpina* genome.

To address the function of *AaMIR157A* and *AaMIR157C* genes during floral transition, CRISPR-Cas9 reverse genetics was used to recover mutant alleles (see Material and Methods; S8 Fig). Analysis of the resulting mutants showed that *Aamir157a* did not significantly influence flowering, but all of the genotypes carrying *Aamir157c* mutations exhibited early flowering from both the main shoot and lateral branches (Fig 6D, E). Apart from accelerated flowering, *Aamir157c* mutations caused reductions in total leaf number, implicating *AaMIR157C* in the regulation of shoot architecture and reproductive timing. Transcriptional reporter lines were constructed to characterize the spatial expression pattern of *AaMIR157C*. Expression of *AaMIR157C*::*mScarlet* reporter was not detected in the *A. alpina* SAM (Fig 6F), similar to what was described for the *MIR157C::VENUS:GUS* reporter in Arabidopsis (Figs 5D and 6G); however, it was broadly expressed in the epidermis of hypocotyls (Fig 6F) similar to in Arabidopsis (Fig 6G). The genetic and expression data suggest a conserved role for *AaMIR157C* in repressing flowering, illustrating the functional conservation of this paralogue with Arabidopsis *MIR157C*.

## Redundancy between *SPL15* and *FT TSF* in promoting the vegetative to I1 transition

The miR157–SPL15 pathway is most important for flowering under SDs (Fig 3A, B, E and S2 Table; 21). These results suggest that the major effectors of the photoperiodic pathway, FT TSF, overcome the requirement for SPL15 under LDs by promoting extremely early flowering. In support of this, *spl15–1* mutations strongly enhanced the late-flowering phenotype of *ft-10 tsf-1* double mutants under LDs [39] (Fig 7A). Therefore, we investigated the interaction between SPL15 and FT

2

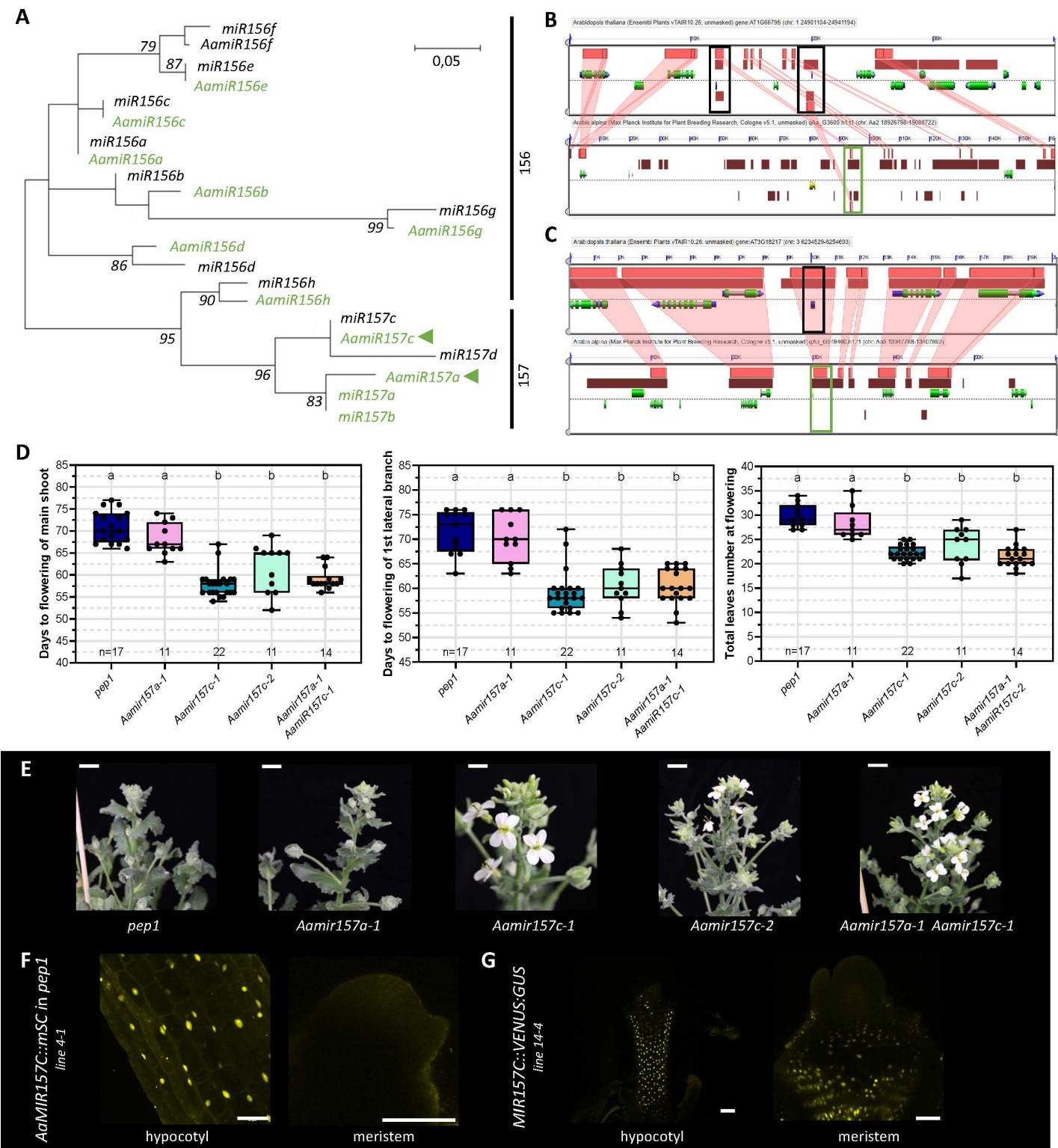

**Fig 6. AamiR157c exhibits similar behaviour to miR157c in regulating flowering in *Arabis alpina*.** (A) Phylogenetic tree of miR156/7 sequences from Arabidopsis (black) and *Arabis alpina* (green). The tree was constructed using the Maximum Likelihood method and bootstrapped with 1000 replicates. The scale bar indicates the branch length that corresponds to the number of substitutions per nucleotide. (B, C) High-resolution analysis of

conservation of synteny in genomic regions from Arabidopsis (top) and *A. alpina* (bottom) containing (B) MIR157A/B and (C) MIR157C genomic regions. The black boxes indicate MIR157A/B/C genes in *A. thaliana*, and the green boxes indicate homologous regions in *A. alpina*. Each panel represents a genomic region, with the dashed line separating top and bottom strands of DNA. Coloured arrows in green/yellow represent protein coding sequences, blue represents transcribed regions, and the full extent of the gene is shown in grey. For each pairwise comparison, light pink boxes represent regions of sequence similarity, with putatively homologous sequences being connected by red shading. (D) Flowering time and total leaf number of the indicated genotypes under long days (LDs). Letters indicate significant differences calculated with ANOVA and Tukey's Honest Significant Difference test; $p < 0.05$. (E) Flowering phenotypes of the indicated genotypes of plants grown for 60LD. Scale bar = 1 cm. (F) Confocal images of *A. alpina* AaMIR157C::mSc at 4–5 weeks and Arabidopsis *MIR157C::VENUS:GUS* plants grown for 9LDs. Scale bar = 100 μm.

TSF in more depth by phenotypically scoring flowering time and inflorescence development of *spl15–1 ft-10 tsf-1* under LDs and SDs.

Under LDs, *spl15–1* did not affect bolting time or rosette leaf number of Col-0, but greatly enhanced the increased bolting time and rosette leaf number of *ft-10 tsf-1*, confirming that SPL15 and FT TSF redundantly regulate floral transition under these conditions (Fig 7A, C and S2 Table). Moreover, *ft-10 tsf-1* formed about four times more cauline leaves than Col-0, whereas *spl15–1* formed the same number as Col-0, and *spl15–1 ft-10 tsf-1* formed a similar number to *ft-10 tsf-1* (4.42 for Col-0; 16.2 for *ft-10 tsf-1*; 15.96 for *spl15–1 ft-10 tsf-1*; Fig 7E and S2 Table). Therefore, under LDs the effect of SPL15 on the transition from vegetative phase to I1, in which cauline leaves and axillary inflorescence branches are formed, is redundant with FT TSF, but in the transition from I1 to I2 phase, in which flowers are formed, only FT TSF plays a major role and SPL15 does not affect this transition, even if FT TSF are mutated.

Under SDs, bolting time and rosette leaf number are strongly increased in *spl15–1* mutants compared with Col-0 (Fig 7B, D). *FT TSF* are not transcribed in leaves of Col-0 under SDs, and the effect of *spl15–1* on bolting time is only weakly enhanced by *ft-10 tsf-1*. This result supports the proposal that SPL15 performs the role of initiating the transition from vegetative to I1 phase under SDs, and that FT TSF has little effect even in *spl15–1*. Moreover, both FT TSF and SPL15 relatively weakly promote the I1 to I2 transition, so that *ft-10 tsf-1* and *spl15–1* form, respectively, about 50% and 25% more cauline leaves than Col-0, and the triple mutant has an additive effect, forming approximately double the number of cauline leves (8,91 for Col-0; 12,38 for *ft-10 tsf-1*; 10,86 for *spl15–1*; 16,32 for *spl15–1 ft-10 tsf-1*; Fig 7E, F and S2 Table). These data suggest that FT TSF and SPL15 both promote vegetative to I1 transition, but their relative importance depends on photoperiod, and that particularly under LDs, FT TSF promotes I1 to I2 transition, whereas SPL15 has a weaker or no effect on this transition.

## Discussion

The *MIR156* and *MIR157* gene families are considered to have similar functions in regulating vegetative phase change and to be genetically redundant, but whether miR156 and miR157 have distinct roles in regulating floral transition has not been extensively tested. Here, we demonstrate that *MIR157C* is more important than *MIR156AC* in delaying bolting and flowering under SDs, when SPLs play a critical role in floral transition, and that its function in repressing flowering is conserved in the related perennial species *A. alpina*. Taken together, this study demonstrates the spatial and temporal expression patterns of *MIR156* and *MIR157* paralogues at the shoot apex, identifies the significance of *MIR157C* in delaying flowering under SDs (Fig 8) and in *A. alpina*, and shows that *MIR157C* activity reduces the expression of SPL15, a promoter of internode elongation and flowering, in the SAM and the shoot under SDs.

### Spatial and temporal patterns of expression of *MIR156* and *MIR157* genes

The dominant concept in the regulation of miR156 and miR157 is that their levels progressively fall as the plant ages [2,10–12]. This progressive downregulation is linked to cell division and therefore developmental age of the tissue, and involves accumulation of trimethylation at lysine 27 of histone 3 (H3K27me3) [13,51]. We used TraPR small RNA

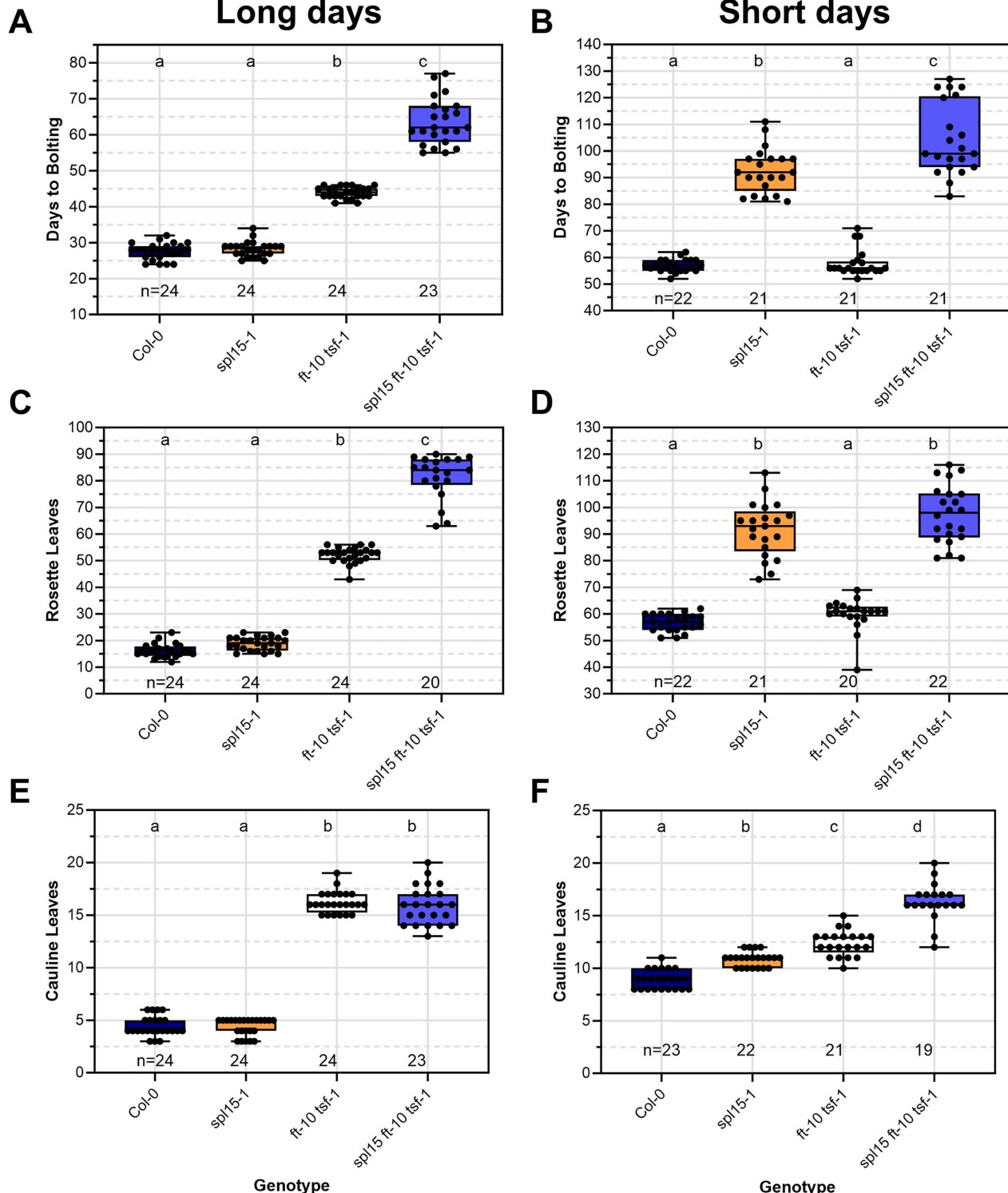

**Fig 7. Genetic interactions between SPL15 and the FT TSF flowering pathway.** (A, B) Bolting time, (C, D) number of rosette leaves and (E, F) number of cauline leaves of the indicated genotypes under (A, C, E) long-day (LD) and (B, D, F) short-day (SD) conditions. Letters indicate significant differences calculated with ANOVA, Tukey's Honest Significant Difference test; $p < 0.05$.

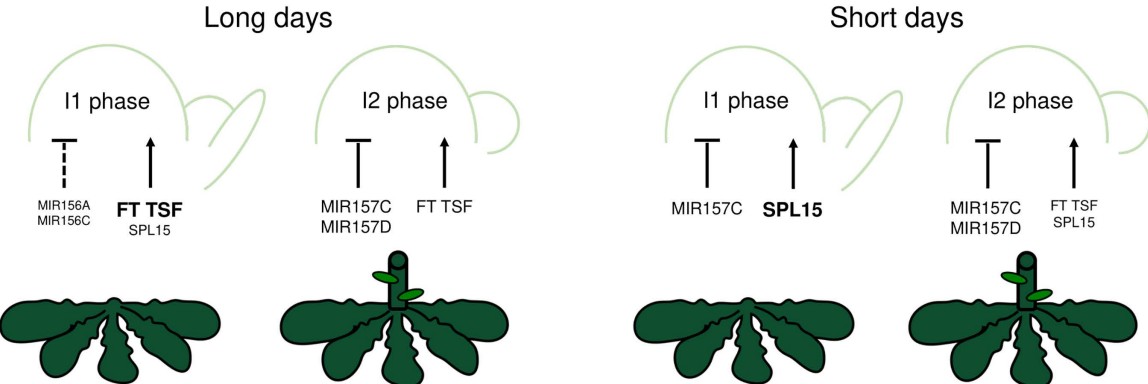

**Fig 8. Schematic representation of regulation of floral transition and inflorescence development by the miR156/miR157–SPL15 module in Arabidopsis.** Left, plants grown under long days. The apex on the left illustrates the contribution of various genes to regulating the transition from vegetative to I1 phase, and the apex on the right illustrates the importance of those genes that regulate the transition from the I1 to I2 phase. Right, plants grown under short days. The apex on the left illustrates the contribution of various genes to regulating the transition from vegetative to I1 phase, and the apex on the right illustrates the importance of those genes that regulate the transition from the I1 to I2 phase. Lines with blunt ends indicate negative regulation, and arrows indicate positive regulation. V, vegetative; I1, I1 phase; I2, I2 phase; FT, FLOWERING LOCUS T; TSF, TWIN SISTER OF FT; SPL15, SQUAMOSA PROMOTER-BINDING PROTEIN-LIKE 15.

sequencing to quantify the levels of miR156 and miR157 within RISCs in shoot apices. In this analysis, levels of miR156 fell rapidly in apices of older plants, whereas levels of miR157 fell more slowly at the same stage and were significantly more abundant than levels of miR156. The relatively high level of miR157 within RISCs in apices of older plants is supported by the higher levels of mature miR157 than miR156 that were found in sRNAs extracted from inflorescences [17]. He et al [9] found a higher level of miR157 than miR156 in 11-day-old seedlings, which contrasts with our description of higher levels of miR156 than miR157 in RISCs in apices of younger plants. This discrepancy might be explained by our use of TraPR for purification of sRNAs, because this method only detects miRNA within RISCs, and miR157 was previously proposed to load less efficiently than miR156 into AGO1 [9].

*MIRNA* paralogues can be expressed in different spatial or temporal patterns. In shoot apices, *MIR156C* transcription was previously detected in the rib zone but not the central or peripheral zones of the SAM from 3 days after germination [17]. We defined the patterns of reporters for *MIR156A*, *MIR156B* and *MIR156C* as well as *MIR157A*, *MIR157C* and *MIR157D* at apices by confocal microscopy. *MIR156B::VENUS:GUS* was expressed throughout the epidermis at the apex, except at the central zone of the SAM from which it was excluded. This pattern did not appear to alter with the age of the plant, and *MIR156B* might therefore modulate SPL activity in the epidermis throughout the life of the plant. In young plants, *MIR156C* was initially detected in the SAM and then later was restricted to the rib region, as previously described [17], before being fully repressed even in apices of older plants. The pattern of expression of *MIR156A* was similar to that of *MIR156C,* but *MIR156A* was not detected in the SAM even in very young plants. Expression of both *MIR156A* and *MIR156C* is almost undetectable throughout the apex by the time flowers are formed under LDs and SDs. However, under LDs, *MIR156C* was still detected in the lower rib region at 11LD and may act in these tissues around that time to weakly delay internode elongation and bolting. These results suggest that *MIR156AC* expression is undetectable at apices during flowering under SDs, when SPL TFs are most important for floral induction.

The spatial patterns of expression of *MIR157* genes have not previously been described at shoot apices. We found that *MIR157C* expression was similar to that of *MIR156AC* in the rib region or upper stem of younger plants, but it then persisted for longer until floral transition at 5wSD or 14LD, at least in the stem. By contrast, *MIR157D* expression was undetectable prior to floral transition and appeared in the axils of cauline leaves and in flowers in the inflorescence. This developmental regulation of *MIR157D* does not follow the age-related reduction shown by *MIR156AC*. *MIR157A*

expression was not detected in the apex, but was present in developing leaf vasculature. Overall, therefore, *MIR156B* and *MIR157D* are developmentally regulated in apices and do not show age-dependent repression, whereas *MIR157C* is slowly reduced in expression with age but persists in the upper stem until flowering. Therefore, among those genes characterized by confocal microscopy, only *MIR156A*C showed the characteristic high-level of expression in juvenile apices and reduction in adult apices that is most associated with these miRNA families.

The eight members of the Arabidopsis *MIR156* family were conserved in genomic location and sequence in *A. alpina*, whereas the *MIR157* family was more variable and only two (*MIR157A* and *MIR157C*) of the four members present in Arabidopsis were also present in *A. alpina*. By contrast, all twelve Arabidopsis genes were conserved in the more closely related *A. lyrata* and *Capsella rubellum* [52]. The presence of mature miR156 was reported in all vascular plants tested [53,54] and in moss [55], but these studies did not distinguish between miR156 and miR157. In addition to miR156, *SPL* mRNAs are targeted by miR529 in mosses and other members of the Embryophyta as well as monocotyledonous species, but this miRNA is absent in core eudicots [53,54]. In grasses, *MIR529* genes are more strongly expressed in the panicle whereas *MIR156* genes are more highly expressed in juvenile vegetative tissues [56–58]. These observations together with our expression analysis of *MIR157* genes in Arabidopsis, suggest that *they* might have evolved more recently than the core *MIR156* genes, be more variable in the Brassicaceae than *MIR156*, and have roles in regulating SPLs in the inflorescence as observed for *MIR529* in panicles of grasses.

### Roles of miR156 and miR157 in stem elongation, flowering and inflorescence development

Under LDs, flowering occurs rapidly through the florigen pathway and is initiated early in plant development when levels of miR156 in RiSCs are still higher than those of miR157. Accordingly, under these conditions, strong reductions in miR156 in the *mir156ac* mutant caused earlier bolting and flowering in some experiments, but *mir157 cd* mutations had no effect on these phenotypes under LDs. A similar result was obtained with vegetative phase change, because leaves formed early during vegetative development showed morphological characteristics of adult leaves in *mir156ac* mutants, but in *mir157 cd* mutants were similar to those of Col-0. This result supports the idea that miR156 is a stronger repressor of vegetative phase change than miR157 [9]. Similarly, *mir156ac* mutants showed the longer plastochron length during vegetative development that is expected for increased *SPL* expression, whereas *mir157 cd* mutants showed a shorter plastochron at this stage. The basis of the shorter plastochron of the latter mutants is not clear, but might be related to the different spatial patterns of expression of the *MIR157* and *MIR156* genes, because expression of *MIR156* genes from heterologous promoters was previously shown to have different effects on plastochron length [46]. Complete loss of miR156 in the hextuple *mir156* mutant and complete loss of miR156 and miR157 in the duodecuple mutant caused extreme early flowering under LDs [17]. These data also suggest that miR156 is more important in flowering of Col-0 plants early in development under LDs than miR157, but that miR157 contributes in *mir156* mutants. In addition, miR157 does affect inflorescence development under LDs, because in the *mir157 cd* mutant the number of cauline leaves was increased under these conditions but was unaffected by *mir156ac*.

Under SDs, the FT florigen pathway is not active and the SPLs are more critical for floral transition [12,15]. Col-0 plants flower later under these conditions, and as the life-cycle is extended, miR157 levels within RiSCs persist at a higher level than those of miR156. Accordingly, we found that the *MIR157 CD* genes were more important than *MIR156* genes in repressing bolting and flowering under SDs, and that they had effects on cauline leaf number and shape that *MIR156* genes did not. In addition, in the smaller *MIR157* family in *A. alpina*, the role of the *MIR157C* orthologue in delaying flowering was conserved.

### Contribution of the SPL15 miR157 module in flowering and inflorescence development

Five *SPL genes* (*SPL3*, *SPL4*, *SPL5*, *SPL9* and *SPL15*) that are negatively regulated by miR156/miR157 have been reported to promote flowering [12,15,27–30]. Mutations in *SPL15* are sufficient to strongly delay flowering under SDs,

whereas transgenes expressing miR156/miR157-resistant versions of *SPL15* mRNA cause early bolting and flowering [15], and genetic epistasis experiments showed that *spl15* mutations delayed bolting of *mir157c* and *mir157 cd* mutants. Moreover, the fluorescence signal of VENUS9A:SPL15 was stronger in the apex of *pSPL15 mir157c-3 mir157d-2* double mutant than in Col-0. These experiments support the idea that miR157 directly targets *SPL15* mRNA, and interestingly, miR157c and mir157d are one nucleotide longer than miR156a or miR156c, and in the case of miR157c this extends the complementarity with *SPL15* mRNA [9]. Furthermore, *spl15 mir157c mir157d* triple mutants bolted much earlier than *spl15* mutants, suggesting that under SDs *MIR157C* delays bolting by repressing other *SPL* genes in addition to *SPL15*. A major contributor to this effect is *SPL4*, because *spl4 spl15 mir157c* bolted significantly later than *spl15 mir157c*. *SPL9*, the paralogue of *SPL15*, and *SPL3* and *SPL5*, paralogues of *SPL4*, might also contribute. *FUL* is bound and activated by several of these SPL family members [12,15,28,59] and *ful* mutations delay flowering under SDs [32,49]. We found that FUL is more highly expressed at the SAM of *mir157 cd* mutants from around 5wSDs, and that *ful* mutations were fully epistatic to *mir157c* and *mir157 cd* in terms of bolting time, so that the higher-order mutants bolted at the same time as *ful* and later than Col-0. Therefore, FUL is required to promote bolting downstream of *MIR157C*, and increased expression of SPLs in *mir157c* does not overcome the requirement for FUL. In *mir157 cd* mutants, the abundance of SPL15 at the SAM and in the cortex of the stem below the SAM increased earlier in development. Nevertheless, at flowering *MIR157C* expression was only detected in the stem and not in the SAM, whereas *MIR157D* expression was only detected in the axils of cauline leaves. Therefore, *MIR157 CD* reduce expression of SPL15 in the SAM and cortex where they are not expressed. This non-autonomy might be explained by movement of the miR157 expressed from *MIR157C* in the lower stem into the upper internodes and SAM, or by *MIR157C* indirectly affecting expression of SPL15 in the SAM via an intermediate mobile signal that is repressed in the lower internodes. The higher level of rSPL15 accumulation in these regions suggest that SPL is directly regulated by movement of miR157; moreover, movement of miR156 through vasculature was previously proposed in potato [60], miR156 and miR157 were detected in the phloem sap of *Brassica napus* [61] and in Arabidopsis movement of miR156 between the SAM and leaf primordia was proposed to influence leaf development [2,62].

Cauline leaf formation occurs during the I1 phase of inflorescence development, and stops on the transition to I2 when flowers are formed. The cauline leaves of *mir157d-2* plants were shorter and wider, consistent with increased SPL activity. This effect was suppressed in the *mir157c-3 mir157d-2* double mutant, perhaps due to miR157 repressing *SPL*s in a different spatial pattern in the cauline leaf leading to suppression of the phenotype. Further explanation of this interaction will require understanding the roles of different SPL transcription factors in controlling cauline leaf shape. In principle, more cauline leaves can be formed if plants transition earlier from vegetative development to I1 or by delayed transition from I1 to I2. Moreover, an increase in cauline leaf number can be caused by an increased rate of organ initiation in I1, for example due to enlargement of the meristem during floral transition [63,64]. Notably, *mir157c* and *mir157d* both caused an increase in cauline leaf number, particularly under SDs and this was enhanced in the double mutant. The different expression patterns of *MIR157C* and *MIR157D* and the increase in cauline leaf number in the double mutant suggest that they might influence cauline leaf number by different mechanisms. The *mir157c* mutant also bolts earlier, as observed for *rSPL15*, suggesting it might cause an increase in cauline leaf number due to premature transition from vegetative to I1. By contrast, *mir157d* does not cause early bolting and is expressed in the axils of cauline leaves, suggesting that *MIR157D* might reduce cauline leaf number by accelerating acquisition of floral meristem identity and transition from I1 to I2. Therefore, *MIR157D* might repress SPLs in the axils of cauline leaves that would otherwise repress acquisition of floral meristem identity. Notably, *SPL9* and *SPL15* were previously proposed to contribute to the repression of *LATERAL SUPPRESSOR* transcription and axillary meristem formation in the axils of cauline leaves [65]. Furthermore, under SDs the *ful* mutation strongly increases cauline leaf number compared with Col-0 [32,49,66], probably by impairing floral meristem identity, and cauline leaf number is further strongly enhanced in the *mir157 cd ful* triple mutant compared with *ful* or *mir157 cd*. These results indicate how different modulators of cauline leaf and axillary branch development can shape inflorescence development and mutually enhance their effects.

In *A. alpina*, the *Aamir157c* mutation caused early flowering. *AaSPL15* has an important role in the promotion of flowering in *A. alpina*, and *rAaSPL15* caused early flowering. Therefore, the early flowering of the *Aamir157c* mutant might at least partly be due to increased activity of AaSPL15 [39]. Our analysis of the *Aamir157c* mutant was performed in the *A. alpina pep1* mutant background, which does not require vernalization to flower [67]. In the *A. alpina* Pajares genotype, which is wild type for *PEP1* and shows an obligate vernalization requirement, the *AamiR156/AaMIR157* families delay responsiveness to vernalization by reducing the activity of SPL15 after vernalization of juvenile plants [37,39]. Our observation that *Aamir157c pep1* mutants are early flowering suggests that *AaMIR157C* is a major contributor to the juvenile flowering response in the *A. alpina* Pajares background.

*SPL15* and *FT TSF* show genetic redundancy in flowering time that is dependent on the daylength to which the plants are exposed. Under LDs, FT TSF strongly promote vegetative to I1 transition and the I1 to I2 transition. Any effect of SPL15 on these characters is almost undetectable under LDs in the presence of FT TSF. However, in the triple *spl15 ft tsf* mutant the redundant role of SPL15 in vegetative to I1 transition is revealed, but this also showed that only FT TSF regulate I1 to I2 transition under LDs. Under SDs, the relative contribution of *FT TSF* and *SPL15* to the vegetative to I1 transition are reversed, so that *spl15* mutants show a greatly increased rosette leaf number, days to bolting and days to flowering, whereas *ft tsf* are similar to Col-0. In transition from I1 to I2, both *spl15* and *ft tsf* cause slight increases in cauline leaf number that are enhanced in the triple mutant. Therefore, the redundancy between *SPL15* and *FT TSF* mainly occurs in the transition from vegetative to I1, and which of these genes are most important depends on daylength. In the transition from I1 to I2, SPL15 generally has a minor role, although this is slightly greater in SDs, whereas FT TSF influences this transition under both conditions but is more important under LDs. In the I1 to I2 transition, FT TSF have been proposed to activate *AP1* in floral primordia and this may not be mediated by SPL15 which is not expressed in primordia [15,68], whereas both pathways activate *FUL* in the SAM [15,35,36]. Other SPLs are also likely to play roles in both pathways as the level of *SPL3*, *SPL4* and *SPL5* are activated in apices by the FT–FD module [29,35,69] and SPL9 is expressed in primordia and is involved in *AP1* activation [12,27]. Additionally, it has been proposed that SPL3, SPL4 or SPL5 forms a transcriptional co-activator complex with FD to activate floral meristem identity genes during the induction flowering under LDs [27,28]. Further elaboration of the common and distinct target genes of FT TSF, SPL15 and related SPLs, as well as their spatio-temporal expression patterns and the contribution of *MIR157 CD* in regulating these will allow more detailed elucidation of the daylength-dependent gene regulatory networks required for floral transition, stem bolting and floral development.

## Materials and methods

### Plant material and growth conditions

Plants were grown on soil in long-day (16 h light/8 h dark) or short-day (8 h light/16 h dark) conditions at 20–22°C with 150–180 µmol m$^{-2}$ s$^{-1}$ light intensity. All genotypes were in the *Arabidopsis thaliana* (L.) ecotype Columbia-0 (Col-0) background. All the *Arabis alpina* genotypes were in a *pep1* background. The *spl15–1* [47], *spl4* [72], *pSPL15::V9A-SPL15* [15], *pSPL15:: V9A:rSPL15* [15], *ful-2* [48], *pFUL::FUL:9AV ful-2* [42], *MIR156A::GFP* [17], *ft-10 tsf-1* [39] and *spl15–1 ft-10 tsf-1* [39] genotypes were published previously. The following genotypes were generated in this study by CRISPR-Cas9 mutagenesis: *miR156a-4, mir156c-4, mir157a-3, mir157b-3, miR157c-3, mir157d-2, Aamir157a-1, Aamir157c-1, Aamir157c-2* and *Aamir157a-1 Aamir157c-1*. To generate *mir157c-3 mir157d-2 pFUL::9AV:FUL ful-2* and *mir157c-3 mir157d-2 pSPL15::9AV:SPL15*, *mir157c-3 mir157d-2* was crossed with the previously published *pFUL::FUL:9AV ful-2* [42] and *pSPL15::V9A-SPL15* [15] and genotyped with the primers listed in S3 Table. The *mir156a-4 mir156c-4 spl15–1, mir157c-3 mir157d-2 spl15–1, mir157c-3 spl4 spl15–1, spl4 spl15–1, mir157c-3 ful-2, mir157d-2 ful-2, mir157c-3 mir157d-2 ful-2, mir156a-4 mir156c-4, mir157c-3 mir157d-2* and *mir157a-3 mir157b-3 mir157c-3* genotypes were obtained by crossing and genotyping with the primers listed in S3 Table. The reporters *MIR156B::VNG, MIR156C::VNG, MIR157A::VNG MIR157C::VNG, MIR157D::VNG* and *AaMIR157C::2HA-mScarlet* were generated in this study and are described below.

## Cloning of *MIR156/MIR157* and *AaMIR157C* reporter constructs

The design of *MIR156B, MIR156C, MIR157A, MIR157C, MIR157D* and *AaMIR157C* reporter lines followed a previously published strategy [32]. Full intergenic regions of *MIR156B, MIR156C, MIR157A, MIR157C* and *MIR157D* were amplified by PCR and cloned into a pDONR entry vector via BP reaction (Invitrogen). Subsequently, the miRNA-encoding stem–loop region was replaced by the coding region of Venus-GUS or 2HA-mScarlet using Gibson Assembly [70]. The entry clones were subcloned via LR reaction (Invitrogen) into the binary vector pEarlyGate301 for *MIR157C::VENUS::GUS* and *AaMIR157C::mSc*, and pBGW.0 for *MIR156B::VENUS::GUS, MIR156C::VENUS:GUS, MIR157A::VENUS:GUS* and *MIR157D::VENUS:GUS*. Next, the plasmids were introduced into *Agrobacterium tumefaciens* strain GV3101 and Col-0 Arabidopsis and *pep1 A. alpina* plants were transformed via floral dipping [71]. Seeds transformed with the bar selection cassette were selected on soil by spraying with 0.001% glufosinate-ammonium. Surviving plants were genotyped via PCR to confirm the presence of the transgene. Several independent lines were generated and analysed, two representative lines were further imaged and presented in the manuscript. Primers used for plasmid construction and genotyping are listed in S3 Table.

## Generation of *MIR156/MIR157* CRISPR-Cas9 alleles

New mutations in Arabidopsis *MIR156* and *MIR157* genes were generated by CRISPR-Cas9. In brief, sgRNAs were designed that targeted the stem–loop region. We took advantage of the high sequence similarity between the precursor genes to design multi-target sgRNAs (S3 Table). We designed one sgRNA to target either *MIR156A, MIR156C* or *MIR156D* individually, or, *MIR157A* and *MIR157B, or MIR157C* and *MIR157D*. The U6-26 promoter-sgRNA cassette was cloned into the pGreen0229 vector as described in [72]. The plasmids were introduced into *A. tumefaciens* strain GV3101 and floral dipping was used to transform wild-type Arabidopsis plants [71]. T1 transgenic plants harbouring the sgRNA-Cas9 construct were selected on soil for the presence of the bar gene. CRISPR-Cas9-free T2 and T3 plants containing mutations were selected via PCR. The mutations were identified with a combination of PCR and Sanger sequencing (S1B Fig). The *mir157a-3 mir157b-3* and *mir157c-3 mir157d-2* mutants each arose from double editing events within the same plant. The *mir157c-3* and *mir157d-2* mutants were obtained via backcrossing.

The *A. alpina* CRISPR-Cas9 mutants were generated using a similar strategy: the U6-26 promoter-sgRNA cassette was cloned into the pHEE401, which allows the incorporation of two specific-target sgRNAs. The resulting construct, which targeted both *AaMIR157A* and *AaMIR157C*, was transformed into *pep1* mutant plants by the Agrobacterium-mediated floral dip method [39]. Primers used for plasmid construction and genotyping are listed in S3 Table.

## DNA extraction and genotyping

For genotyping, genomic DNA (gDNA) was extracted using the BioSprint 96 DNA Plant Kit (Qiagen) in combination with the Biosprint automation platform according to the manufacturer's instructions. The extracted gDNA was eluted to a final volume of 100 µL with ddH$_2$O. GoTaq DNA Polymerase and GoGreen Buffer (Promega GmbH) were used for all genotyping assays. All primers used for genotyping are listed in S3 Table.

## Confocal imaging

Shoot apices at different developmental stages were dissected under a stereo microscope and fixed as described in [63]. Tissue was cleared in ClearSee [73] for 2–5 days, depending on the developmental stage. Cell walls were stained with SCRI 2200 Renaissance [74] (0.1% in ClearSee) overnight at room temperature. *A. alpina* samples were imaged as live tissue. Confocal images were acquired with a TPS SP8 confocal microscope (Leica Microsystems GmbH). The excitation wavelength was 405 nm for Renaissance, 488 nm for GFP, 514 nm for VENUS, 561 nm for mScarlet. Image collection was performed at 430–470 nm for Renaissance, 500–520 nm for GFP, 520–534 nm for VENUS and 580–620 nm for mScarlet.

Images were processed in ImageJ, which involved applying consistent brightness settings and generating maximum intensity projections of 10-µm optical sections at the meristem centre. Imaging and processing parameters for fluorescent proteins were determined for the first time point and remained constant throughout the experiment, unless stated otherwise.

### RNA extraction and RT-qPCR

Plant tissue was harvested at the indicated times and immediately snap frozen in liquid nitrogen. Total RNA was isolated using the miRNeasy Tissue/Cells Advanced Mini Kit (Qiagen) according to the manufacturer's instructions. The RNA was treated with Ambion DNase I (Thermo Fisher Scientific Inc) and 1 µg total RNA was used for cDNA synthesis with the Superscript IV first-strand synthesis system (Thermo Fischer Scientific Inc). Transcript levels were quantified by quantitative real-time PCR with iQ SYBR green supermix (Bio-Rad) in a CFX384 Touch Real-Time PCR Detection system (Bio-Rad). Three technical replicates were performed for three independent biological triplicates, and expression of *PEROXIN4* (*PEX4*) and *PP2A* (*PROTEIN PHOSPHATASE 2A*) genes was used to normalize the data. Primers used for expression analyses are listed in S3 Table.

### Small RNA extraction and data processing

Apices were harvested in three biological replicates for all the indicated time points. Small RNAs were extracted with the TraPR Small RNA Isolation Kit (Lexogen). Sample extraction and library preparation were carried out at the Max Planck Genome Centre Cologne. Circa 10 million single-end 150-bp reads were sequenced per sample using the Illumina HiSeq 3500 system. Adapter sequences were trimmed from raw reads using Cutadapt version 3.4. The parameters were selected for a minimum read length of 17 nucleotides and a minimum Illumina quality score of 20. Trimmed sequence reads were analysed with DeconSeq version 0.4.3 to filter out non-plant contaminations [75]. Mapping of the sRNA reads to the Arabidopsis genome was performed with the Manatee tool, version 1.2 [76] and used annotations from the Araport11 release [77] supplemented with the miRbase 22 release [78]. Annotations were limited to miRNAs, short-interfering RNAs and other non-coding RNAs. Other annotations, including protein-coding genes, were excluded from the analysis.

### Phylogenetic analysis

Sequences similar to Arabidopsis miR156 and miR157 were obtained from *A. alpina* genome database (https://ucsc-browser.mpipz.mpg.de) through BLAT search using default parameters. Sequence alignments were performed by ClustalW [79], and Gblocks [80] was used to remove less conservative regions. A phylogenetic tree was elaborated using Maximum Likelihood method (Bootstrapped with 1000 replicates). Alignment and phylogeny were carried out in MEGA version X.

### Microsynteny analysis

Microsynteny analysis was performed using the Genome Evolution Analysis (GEvo) tools of the Comparative Genomics Platform (CoGe) [81]. Briefly, similar regions between the Arabidopsis and *A. alpina* genomes were searched using CoGeBlast and then further examined using GEvo. We used the default setting to define the minimum number of collinear genes for two regions to be called syntenic.

### Statistical analysis

Data were analysed using R version 4.0.3 and associated statistical packages [82] and GraphPad Prism 9. Statistical differences were determined with one- or two-factor ANOVA comparisons. For clearly non-parametric data, a Kruskal-Wallis (KW) omnibus test was applied. If significant differences were identified, post-hoc multiple comparisons were performed. Tukey's HSD multiple comparison test was used to determine differences identified by ANOVA (both from base R) and

a Nemenyi post-hoc test (PMCMRplus package) was used to analyse differences after the Kruskal-Wallis test (base R). Levene's test of equal variance was employed to determine differences in sample variation. For the box plots, the boxes represent the 25% and 75% quartiles. The horizontal line inside the box indicates the median, and each point represents an individual plant.

## Supporting information

**S1 Table. Read count of miR156/miR157 in shoot apices of plants grown under long days (LDs) and short days (SDs).**
(XLSX)

**S2 Table. Numerical data that underlie graphs.**
(XLSX)

**S3 Table. List of primers used in this study.**
(XLSX)

**S1 Fig. (A) Abundance of mRNA transcripts of all members of the MIR156/157 family in apices during floral transition under LDs extracted from RNAseq data.** TPM denotes transcripts per kilobase of exon per million mapped fragments. Data retrieved from Cerise et al. 2023. (B) Sequence of the Arabidopsis CRISPR-Cas9 alleles generated in this study. All deletions affect the conserved hairpin sequence that is required for the correct biogenesis of miR156 and miR157. For each isoform, the wild-type reference sequence is shown above and the mutant sequence is shown below. The miRNA and miRNA* sequences are highlighted in blue. Red letters indicate non-template insertions.
(TIFF)

**S2 Fig. mir156/mir157 family members regulate vegetative traits.** (A) Rosette phenotype of Col-0, mir156a-4 mir156c-4, mir157c-3 mir157d-2, mir157a-3 mir157b-3 mir157c-3 and rSPL15 plants grown for four weeks in short days (4wSD). (B) Morphology of fully expanded rosette leaves of wild-type plants and mutants for genes encoding miR156, miR157, and rSPL15 at 4wSD. (C) Distribution of the divergence angle between successive rosette leaves in 5wSD plants. The horizontal line indicates the ideal angle of 137.5° and dashed lines show the minimum and maximum angles measured for wild type. Letters indicate statistically significant differences in variation as calculated by Levene's and Tukey's post-hoc HSD tests.
(TIFF)

**S3 Fig. (A) Expression levels of endogenous *MIR156A* and *MIR156C* genes in apices of Col-0 plants grown in long days (LD).** (B) Expression levels of endogenous *MIR157A* and *MIR157C* genes in apices of Col-0 plants grown in short days (SD).
(TIFF)

**S4 Fig. Characterization of *MIR156A* and *MIR156C* expression patterns.** Confocal images showing the expression at the shoot apical meristem of (A) *MIR156A::GFP* and (B) *MIR156C::VENUS:GUS* in plants grown short days (SD). Fluorescence from the Venus protein is artificially coloured in yellow, fluorescence from the GFP protein is artificially coloured in green. Scale bars = 50 μm.
(TIFF)

**S5 Fig. Characterization of *MIR156B* expression pattern.** Confocal images showing the expression at the shoot apical meristem of *MIR156B::VENUS:GUS* in short days (SD). Fluorescence from the Venus protein is artificially coloured in yellow, and the fluorescence from the Renaissance dye is artificially coloured in grey. Scale bar = 50 μM.
(TIFF)

**S6 Fig. Characterization of *MIR157A* expression pattern.** Confocal images showing the expression of *MIR157A::VE-NUS:GUS* in short days (SD). Fluorescence from the Venus protein is artificially coloured in yellow, and the fluorescence from the Renaissance dye is artificially coloured in grey. Scale bar = 50 μM. CL, Cauline leaf.
(TIFF)

**S7 Fig. Characterization of *MIR157C* and *MIR157D* expression patterns.** Confocal images showing the expression *MIR157C::VENUS:GUS* in (A) short days (SD), (B) long days (LD) and (C) *MIR157D::VENUS:GUS* in long days (LD). Fluorescence from the Venus protein is artificially coloured in yellow, and the fluorescence from the Renaissance dye is artificially coloured in grey. CL, Cauline leaf. Scale bar = 50 μM.
(TIFF)

**S8 Fig. Sequence of the *Arabis alpina* CRISPR-Cas9 alleles generated in this study.** All deletions affect the conserved hairpin sequence that is required for the correct biogenesis of *AamiR157*. For each isoform, the wild-type reference sequence is shown above and the mutant sequence is shown below. The miRNA and miRNA* sequences are highlighted in red (AamiR157c) and blue (AamiR157a). Black letters indicate miRNA precursor and green letters indicate the genomic region.
(TIFF)

## Acknowledgments

We thank the staff of the Plant Cultivation Facilities at the MPIPZ for assistance with plant growth. We thank Ton Timmers from the Central Microscopy service (MPIPZ), the Max Planck Genome Centre Cologne and the members of the Coupland group, especially Enric Bertran Garcia de Olalla, Pedro de los Reyes and Sheila Gomez-Colliga, for their helpful assistance and fruitful discussions. We thank Jiawei Wang for providing seeds of the *MIR156A:GFP* transgenic line.

## Author contributions

**Conceptualization:** Adrian Roggen, George Coupland.

**Data curation:** Adrian Roggen, Alba Lloret, Yohanna Miotto.

**Funding acquisition:** Alba Lloret, Yohanna Miotto, George Coupland.

**Investigation:** Adrian Roggen, Alba Lloret, Yohanna Miotto, Kang Wang, Kerstin Luxa, Vidya Oruganti, Serena Della Pina, Annabel D. van Driel, Youbong Hyun, Bruno Huettel.

**Resources:** George Coupland.

**Supervision:** George Coupland.

**Visualization:** Adrian Roggen, Alba Lloret, Yohanna Miotto.

**Writing – original draft:** Adrian Roggen, George Coupland.

**Writing – review & editing:** Yohanna Miotto, George Coupland.

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
