## [Decision Letter · Decision Letter 0]

9 Apr 2025

PGENETICS-D-25-00166

The MIR157–SPL15 module regulates flowering and inflorescence development in Arabidopsis thaliana under short days and in Arabis alpina

PLOS Genetics

Dear Dr. Coupland,

Thank you for submitting your manuscript to PLOS Genetics. After careful consideration, we feel that it has merit but does not fully meet PLOS Genetics's publication criteria as it currently stands. Therefore, we invite you to submit a revised version of the manuscript that addresses the points raised during the review process.

Please submit your revised manuscript within 60 days (June 8). If you will need more time than this to complete your revisions, please reply to this message or contact the journal office at plosgenetics@plos.org. Please include the following items when submitting your revised manuscript:

We look forward to receiving your revised manuscript.

Kind regards,

Mathilde Grelon

Academic Editor

PLOS Genetics

Anne Goriely

Editor-in-Chief

PLOS Genetics

Aimée Dudley

Editor-in-Chief

PLOS Genetics

Anne Goriely

Editor-in-Chief

PLOS Genetics

**Reviewers' comments:**

Reviewer's Responses to Questions

**Comments to the Authors:**

Reviewer #1: In the manuscript ‘The MIR157–SPL15 module regulates flowering and inflorescence development in Arabidopsis thaliana under short days and in Arabis alpina, Roggen et al. present a detailed study on the contributions of miR156 and miR157 to different flowering-related phenotypes. While the role of miR156 in the regulation of the reproductive transition has been very well investigated, it was not yet clear to what extent MIR157 genes contribute to flowering-related processes. Roggen et al. show using small RNA-seq that RISC-loaded miR157 is not rapidly declining a few weeks after germination, but remains moderately present also later during development, in line with the expression dynamics of MIR156A/C and MIR157 C/D. Interestingly, they show that this role of MIR157C is conserved in the perennial Arabis alpina, where mutation of MIR157C results in early flowering. Using genetic approaches, Roggen et al. show that the activity of miR157 depends on SPL15 (like that of miR156) and that the miR157-SPL15 module plays a role when the FT pathway is not induced (i.e. under short days). The results are clearly presented and the manuscript has been written very well.

We (senior and junior reviewers) only have a few minor comments/suggestions:

- It is not entirely clear why different mutant combinations have been generated. It would be good to add a graph in the Suppl. data with the expression levels for the different MIR156 and MIR157 genes to support for example the selection of MIR156A/C. And explain why mutants for mir157c/d and mir157a/b/c were generated.

- For better readability of the graphs, it would be nice to indicate when the I1 and I2 phases start (in Figures 1 and 2).

- In several of the line graphs, it is very difficult to see what is significant and what not. In Figure 3, the authors even refer to the Suppl. data for this, while it these are relevant data. If they change the graphs to bar graphs in some cases, it will be much easier to indicate significance by using letters.

- Figure 2: Why was the mir157a/b/c mutant not included here?

- Figure 2C: there is no letter c to indicate significance.

- There is a rather abrupt introduction of FUL/SPL15 in the text (lines 250-259) relating to the cauline leave number phenotype. The involvement of FUL here makes it rather complex, and there is no good explanation, nor is it further investigated (e.g. by testing the expression of FUL). The conclusion lines that follow could also ignoring the FUL/SPL15 experiment. It is probably better to delete the experiment on FUL(/SPL15) here (or to work it out better).

- Lines 289 – 291. ‘The expression of the floral meristem identity gene….., and FUL is activated by SPLs.’ The reasoning of this sentence is not very clear (i.e. there is no logical connection between the first and second part of the sentence).

- Line 316: ‘similar time to’ should be ‘similar time as’.

- Lines 302 – 305 and 308-310: it seems that FUL and SPL15 are introduced here, while they have already been introduced in lines 250-259. According to our suggestion, you could leave out lines 250-259. However, if these are retained, the text should be adjusted so to timely introduce FUL and SPL15.

- Please comment on how the reporter lines in the different mutant backgrounds were generated. Were the constructs transformed in the different mutant backgrounds and thus all representing independent construct inserts in the genome? In that case, it is important to mention how many independent lines per construct were assessed.

- Please also indicate in one of the pictures where Rib zone, PZ and CZ can be found.

Reviewer #2: The MIR157–SPL15 module regulates flowering and inflorescence development in Arabidopsis thaliana under short days and in Arabis alpina (PGENETICS-D-25-00166)

The manuscript by Roggen et al. examines the molecular mechanisms underlying the roles of the miR157-SPL15 in flowering regulation under short days in both Arabidopsis thaliana and Arabis alpina. Although it is well-known that the miR156-SPL module functions during vegetative phase changing and flowering time, the role of miR157 in these developmental contexts is less clear. In general, the manuscript is well written and the data are clear and the experiments well planned and executed. However, I have a few questions that may improve the manuscript:

1- In previous publication (He et al., 2018 PloS Genetics), it was shown that miR157 is more abundant than miR156 but has a smaller effect on SPL gene expression than miR156. This may be attributable to the inefficiency with which miR157 is loaded onto AGO1, as well as to the presence of an extra nucleotide at the 5' end of miR157 that is mis-paired in the miR157:SPL duplex. Although the authors showed that miR157 transcripts persisted longer within RISCs late in development (inflorescence development), they did not test the effect of the extra nucleotide in the efficiency of the miRNA:mRNA duplex formation.

Thus, it is difficult to assess whether miR157 effectively regulates SPLs in inflorescence development. Unless I missed it, the manuscript does not provide clear data demonstrating this regulation. If such data are available, it would be helpful to highlight them more explicitly; otherwise, additional experiments could strengthen this conclusion.

2- Neither miR156 nor miR157 mutations seem to affect rosette leaf number in 4wSD plants when comparing with Col-0 and rSPL15 (Fig. 2B). Is there a possible explanation for these phenotypes? The authors showed that the developmental regulation of MIR157D does not follow the general age-associated reduction shown by MIR156 genes (lines 533-534), which suggests that miR157 function is more prominent in adult plants. However, miR157 does accumulate at higher levels in seedlings (He et al., 2018 PloS Genetics). Thus, while miR157 does not affect leaf shape (Supp. Fig. S2), it may have an effect on the plastochron of juvenile plants. Did the authors check that?

3- SPL9 has been shown to promote flowering and it is the primary SPL15 paralogue. The authors mentioned that spl15 mir157c mir157d triple mutants still bolted much earlier than spl15 mutants (lines 583-584), which indicates that other SPLs may be at play. Are the mutations in both paralogues (SPL9 and SPL15) sufficient to suppress mir157 mutant flowering phenotypes? Have the authors examined spl9 spl15 mir157 mutants? If not, this would be an informative experiment to determine whether SPL9 and SPL15 together are the primary contributors to floral transition under SD, or if additional SPL genes are involved.

Reviewer #3: The Manuscript is well written and all data is supported by the figures and tablles. Here are some minor questions that:

1. The LWR of cauline leave as shown in Figure 2 is lower in miR157d-2 but higher in the double mutant of miR157d-2 miR157c-3, how?

2. In figure 5A, why no real-time PCR was done on miR156 in long days?

**Have all data underlying the figures and results presented in the manuscript been provided?**

Reviewer #1: Yes

Reviewer #2: Yes

Reviewer #3: Yes

PLOS authors have the option to publish the peer review history of their article (what does this mean? ). If published, this will include your full peer review and any attached files.

**Do you want your identity to be public for this peer review?** For information about this choice, including consent withdrawal, please see our Privacy Policy .

Reviewer #1: No

Reviewer #2: **Yes: ** Fabio Nogueira

Reviewer #3: **Yes: ** Sachin Teotia

**Figure resubmission:**
---

## [Decision Letter · Decision Letter 1]

8 Jul 2025

Dear Dr Miotto,

We are pleased to inform you that your manuscript entitled "The MIR157–SPL15 module regulates flowering and inflorescence development in Arabidopsis thaliana under short days and in Arabis alpina" has been editorially accepted for publication in PLOS Genetics. Congratulations!

Yours sincerely,

Mathilde Grelon

Academic Editor

PLOS Genetics

Anne Goriely

Editor-in-Chief

PLOS Genetics

Aimée Dudley

Editor-in-Chief

PLOS Genetics

Anne Goriely

Editor-in-Chief

PLOS Genetics

Comments from the reviewers (if applicable):

Reviewer's Responses to Questions

**Comments to the Authors:**

Reviewer #1: We appreciate the work that the authors did on the revision of the paper. They addressed all our points satisfactorily.

We encountered three other minor things:

The choice for the different MIR156/157 genes is clearer in the Results section now. However, it is confusing that in the introduction, the authors refer to MIR156A/C and MIR157A/C as together representing 90% of the transcripts (in seedlings). From the introduction, one can then take that these are the four most important ones in general, while in flowering time regulation, MIR157D is much more important. Therefore, we advise to remove these lines from the introduction.

VENUS is used as a reporter, but the notation of the VENUS-containing constructs is not always clear. For example, in Materials & Methods, Lines 723-724, it is referred to as VNG (instead of VENUS::GUS). In Figure 4, V9A is used, which stands for Venus-9-Alinina, but this term is then only introduced in the legend. For clarity, it would be better to just write VENUS in the Results and only clarify in the Materials & Methods specific aspects.

- Leves instead of leaves in line 503

Reviewer #2: The authors have satisfactorily addressed all of my concerns.

Reviewer #3: The authors have satisfactorily addressed the comments.

**Have all data underlying the figures and results presented in the manuscript been provided?**

Reviewer #1: Yes

Reviewer #2: Yes

Reviewer #3: Yes

PLOS authors have the option to publish the peer review history of their article (what does this mean? ). If published, this will include your full peer review and any attached files.

**Do you want your identity to be public for this peer review?** For information about this choice, including consent withdrawal, please see our Privacy Policy .

Reviewer #1: No

Reviewer #2: **Yes: ** Fabio Nogueira

Reviewer #3: **Yes: **

**Data Deposition**

http://datadryad.org/submit?journalID=pgenetics&manu=PGENETICS-D-25-00166R1

**Press Queries**

---

## [Editor Report · Acceptance letter]

PGENETICS-D-25-00166R1

The MIR157–SPL15 module regulates flowering and inflorescence development in Arabidopsis thaliana under short days and in Arabis alpina

Dear Dr Coupland,

We are pleased to inform you that your manuscript entitled " 

The MIR157–SPL15 module regulates flowering and inflorescence development in Arabidopsis thaliana under short days and in Arabis alpina" has been formally accepted for publication in PLOS Genetics! Your manuscript is now with our production department and you will be notified of the publication date in due course.

With kind regards,

Anita Estes

PLOS Genetics

On behalf of:
